# Parameter Prediction for Unseen Deep Architectures

**Boris Knyazev**[1,2][*]  **Michal Drozdzal**[4,†]   **Graham W. Taylor**[1,2,3,†]   **Adriana Romero-Soriano**[4,5,†]

[1] University of Guelph   [2] Vector Institute for Artificial Intelligence
[3] Canada CIFAR AI Chair   [4] Facebook AI Research   [5] McGill University
[†]equal advising

https://github.com/facebookresearch/ppuda

## Abstract

Deep learning has been successful in automating the design of features in machine learning pipelines. However, the algorithms optimizing neural network parameters remain largely hand-designed and computationally inefficient. We study if we can use deep learning to directly predict these parameters by exploiting the past knowledge of training other networks. We introduce a large-scale dataset of diverse computational graphs of neural architectures – DEEPNETS-1M– and use it to explore parameter prediction on CIFAR-10 and ImageNet. By leveraging advances in graph neural networks, we propose a hypernetwork that can predict performant parameters in a *single forward pass* taking a fraction of a second, even on a CPU. The proposed model achieves surprisingly good performance on *unseen* and *diverse* networks. For example, it is able to predict all 24 million parameters of a ResNet-50 achieving a 60% accuracy on CIFAR-10. On ImageNet, top-5 accuracy of some of our networks approaches 50%. Our task along with the model and results can potentially lead to a new, more computationally efficient paradigm of training networks. Our model also learns a strong representation of neural architectures enabling their analysis.

## 1  Introduction

Consider the problem of training deep neural networks on large annotated datasets, such as ImageNet [1]. This problem can be formalized as finding optimal parameters for a given neural network $a$, parameterized by $\mathbf{w}$, w.r.t. a loss function $\mathcal{L}$ on the dataset $\mathcal{D} = \{(\mathbf{x}_i, y_i)\}_{i=1}^{N}$ of inputs $\mathbf{x}_i$ and targets $y_i$:

$$\arg\min_{\mathbf{w}} \sum\nolimits_{i=1}^{N} \mathcal{L}(f(\mathbf{x}_i; a, \mathbf{w}), y_i), \tag{1}$$

where $f(\mathbf{x}_i; a, \mathbf{w})$ represents a forward pass. Equation 1 is usually minimized by iterative optimization algorithms – e.g. SGD [2] and Adam [3] – that converge to performant parameters $\mathbf{w}_p$ of the architecture $a$. Despite the progress in improving the training speed and convergence [4–7], obtaining $\mathbf{w}_p$ remains a bottleneck in large-scale machine learning pipelines. For example, training a ResNet-50 [8] on ImageNet can take many GPU hours [9]. With the ever growing size of networks [10] and necessity of training the networks repeatedly (e.g. for hyperparameter or architecture search), the classical process of obtaining $\mathbf{w}_p$ is becoming computationally unsustainable [11–13].

**A new parameter prediction task.** When optimizing the parameters for a *new* architecture $a$, typical optimizers disregard past experience gained by optimizing other nets. However, leveraging this past experience can be the key to reduce the reliance on iterative optimization and, hence the high computational demands. To progress in that direction, we propose a new task where iterative optimization is replaced with a *single forward pass* of a hypernetwork [14] $H_{\mathcal{D}}$. To tackle the task,

---

[*]Part of the work was done while interning at Facebook AI Research.

$H_{\mathcal{D}}$ is expected to leverage the knowledge of how to optimize *other* networks $\mathcal{F}$. Formally, the task is to predict the parameters of an *unseen* architecture $a \notin \mathcal{F}$ using $H_{\mathcal{D}}$, parameterized by $\theta_p$: $\hat{\mathbf{w}}_p = H_{\mathcal{D}}(a; \theta_p)$. The task is constrained to a dataset $\mathcal{D}$, so $\hat{\mathbf{w}}_p$ are the predicted parameters for which the test set performance of $f(\mathbf{x}; a, \hat{\mathbf{w}}_p)$ is similar to the one of $f(\mathbf{x}; a, \mathbf{w}_p)$. For example, we consider CIFAR-10 [15] and ImageNet image classification datasets $\mathcal{D}$, where the test set performance is classification accuracy on test images.

**Approaching our task.** A straightforward approach to expose $H_{\mathcal{D}}$ to the knowledge of how to optimize other networks is to train it on a large training set of $\{(a_i, \mathbf{w}_{p,i})\}$ pairs, however, that is prohibitive[2]. Instead, we follow the bi-level optimization paradigm common in meta-learning [16–18], but rather than iterating over $M$ tasks, we iterate over $M$ training architectures $\mathcal{F} = \{a_i\}_{i=1}^M$:

$$\arg\min_\theta \sum_{j=1}^N \sum_{i=1}^M \mathcal{L}\Big(f\Big(\mathbf{x}_j; a_i, H_{\mathcal{D}}(a_i; \theta)\Big), y_j\Big). \tag{2}$$

By optimizing Equation 2, the hypernetwork $H_{\mathcal{D}}$ gradually gains knowledge of how to predict performant parameters for training architectures. It can then leverage this knowledge at test time – when predicting parameters for *unseen* architectures. To approach the problem in Equation 2, we need to design the network space $\mathcal{F}$ and $H_{\mathcal{D}}$. For $\mathcal{F}$, we rely on the previous design spaces for neural architectures [19] that we extend in two ways: the ability to sample distinct architectures and an expanded design space that includes diverse architectures, such as ResNets and Visual Transformers [20]. Such architectures can be fully described in the form of computational graphs (Fig. 1). So, to design the hypernetwork $H_{\mathcal{D}}$, we rely on recent advances in machine learning on graph-structured data [21–24]. In particular, we build on the Graph HyperNetworks method (GHNs) [24] that also optimizes Equation 2. However, GHNs do not aim to predict large-scale performant parameters as we do in this work, which motivates us to improve on their approach.

By designing our diverse space $\mathcal{F}$ and improving on GHNs, we boost the accuracy achieved by the predicted parameters on *unseen* architectures to 77% (top-1) and 48% (top-5) on CIFAR-10 [15] and ImageNet [1], respectively. Surprisingly, our GHN shows good out-of-distribution generalization and predicts good parameters for architectures that are much larger and deeper compared to the ones seen in training. For example, we can predict all 24 million parameters of ResNet-50 in less than a second either on a GPU or CPU achieving $\sim$60% on CIFAR-10 without any gradient updates (Fig 1, (b)).

Overall, our framework and results pave the road toward a new and significantly more efficient paradigm for training networks. Our **contributions** are as follows: (**a**) we introduce the novel task of predicting performant parameters for diverse feedforward neural networks with a single hypernetwork forward pass; (**b**) we introduce DEEPNETS-1M – a standardized benchmark with in-distribution and out-of-distribution architectures to track progress on the task (§ 3); (**c**) we define several baselines and propose a GHN model (§ 4) that performs surprisingly well on CIFAR-10 and ImageNet (§ 5.1); (**d**) we show that our model learns a strong representation of neural network architectures (§ 5.2), and our model is useful for initializing neural networks (§ 5.3). Our DEEPNETS-1M dataset, trained GHNs and code is available at `https://github.com/facebookresearch/ppuda`.

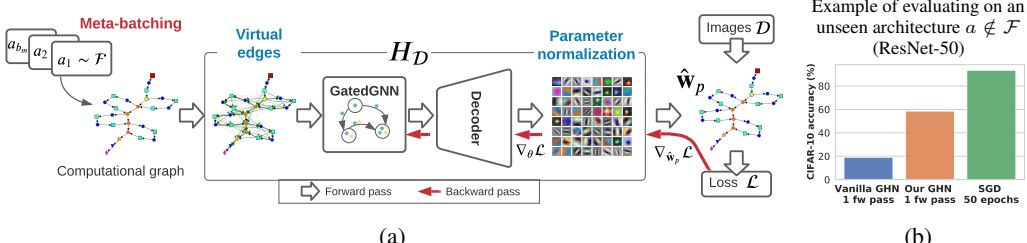

Figure 1: (**a**) Overview of our GHN model (§ 4) trained by backpropagation through the predicted parameters ($\hat{\mathbf{w}}_p$) on a given image dataset and our DEEPNETS-1M dataset of architectures. Colored captions show our key improvements to vanilla GHNs (§ 2.2). The red one is used only during training GHNs, while the blue ones are used both at training and testing time. The computational graph of $a_1$ is visualized as described in Table 1. (**b**) Comparing classification accuracies when all the parameters of a ResNet-50 are predicted by GHNs versus when its parameters are trained with SGD (see full results in § 5).

---

[2]Training a single network $a_i$ can take several GPU days and thousands of trained networks may be required.

## 2 Background

We start by providing a brief background about the network design spaces leveraged in the creation of our DEEPNETS-1M dataset of neural architectures described in § 3. We then cover elements of graph hypernetworks that we leverage when designing our specific GHN $H_\mathcal{D}$ in § 4.

### 2.1 Network Design Space of DARTS

DARTS [19] is a differentiable NAS framework. For image classification tasks such as those considered in this work, its networks are defined by four types of building blocks: *stems*, *normal cells*, *reduction cells*, and *classification heads*. Stems are fixed blocks of convolutional operations that process input images. The normal and reduction cells are the main blocks of architectures and are composed of: 3×3 and 5×5 separable convolutions, 3×3 and 5×5 dilated separable convolutions, 3×3 max pooling, 3×3 average pooling, identity and zero (to indicate the absence of connectivity between two operations). Finally, the classification head defines the network output and is built with a global pooling followed by a single fully connected layer.

Typically, DARTS networks have one stem block, 14-20 cells, and one classification head, altogether forming a deep computational graph. The reduction cells, placed only at 1/3 and 2/3 of the total depth, decrease the spatial resolution and increase the channel dimensionality by a factor of 2. Summation and concatenation are used to aggregate outputs from multiple operations within each cell. To make the channel dimensionalities match, 1×1 convolutions are used as needed. All convolutional operations use the ReLU-Conv-Batch Norm (BN) [7] order. Overall, DARTS enables defining strong architectures that combine many principles of manual [25, 8, 26, 27] and automatic [24, 28–33] design of neural architectures. While DARTS learns the optimal task-specific cells, the framework can be modified to permit sampling randomly-structured cells. We leverage this possibility for the DEEPNETS-1M construction in § 3. Please see § A.1 for further details on DARTS.

### 2.2 Graph HyperNetwork: GHN-1

**Representation of architectures.** GHNs [24] directly operate on the computational graph of a neural architecture $a$. Specifically, $a$ is a directed acyclic graph (DAG), where nodes $V = \{v_i\}_{i=1}^{|V|}$ are operations (e.g. convolutions, fully-connected layers, summations, etc.) and their connectivity is described by a binary adjacency matrix $\mathbf{A} \in \{0,1\}^{|V|\times|V|}$. Nodes are further characterized by a matrix of initial node features $\mathbf{H}^0 = [\mathbf{h}_1^0, \mathbf{h}_2^0, ..., \mathbf{h}_{|V|}^0]$, where each $\mathbf{h}_v^0$ is a one-hot vector representing the operation performed by the node. We also use such a one-hot representation for $\mathbf{H}^0$, but in addition encode the shape of parameters associated with nodes as described in detail in § B.1.

**Design of the graph hypernetwork.** In [24], the graph hypernetwork $H_\mathcal{D}$ consists of three key modules. The first module takes the input node features $\mathbf{H}^0$ and transforms them into $d$-dimensional node features $\mathbf{H}^1 \in \mathbb{R}^{|V|\times d}$ through an embedding layer. The second module takes $\mathbf{H}^1$ together with $\mathbf{A}$ and feeds them into a specific variant of the gated graph neural network (GatedGNN) [34]. In particular, their GatedGNN mimics the canonical order $\pi$ of node execution in the forward (fw) and backward (bw) passes through a computational graph. To do so, it sequentially traverses the graph and performs iterative message passing operations and node feature updates as follows:

$$\forall t \in [1, ..., T] : \left[ \forall \pi \in [\text{fw}, \text{bw}] : \left( \forall v \in \pi : \mathbf{m}_v^t = \sum_{u \in \mathcal{N}_v^\pi} \text{MLP}(\mathbf{h}_u^t), \ \mathbf{h}_v^t = \text{GRU}(\mathbf{h}_v^t, \mathbf{m}_v^t) \right) \right], \quad (3)$$

where $T$ denotes the total number of forward-backward passes; $\mathbf{h}_v^t$ corresponds to the features of node $v$ in the $t$-th graph traversal; MLP$(\cdot)$ is a multi-layer perceptron; and GRU$(\cdot)$ is the update function of the Gated Recurrent Unit [35]. In the forward propagation ($\pi = \text{fw}$), $\mathcal{N}_v^\pi$ corresponds to the incoming neighbors of the node defined by $\mathbf{A}$, then in the backward propagation ($\pi = \text{bw}$) it similarly corresponds to the outgoing neighbors of the node. The last module uses the GatedGNN output hidden states $\mathbf{h}_v^T$ to condition a decoder that produces the parameters $\hat{\mathbf{w}}_p^v$ (e.g. convolutional weights) associated with each node. In practice, to handle different parameter dimensionalities per operation type, the output of the hypernetwork is reshaped and sliced according to the shape of parameters in each node. We refer to the model described above as GHN-1 (Fig. 1). Further subtleties of implementing this model in the context of our task are discussed in § B.1.

Table 1: Examples of computational graphs (visualized using NetworkX [44]) in each split and their key statistics, to which we add the average degree and average shortest path length often used to measure local and global graph properties respectively [45, 46]. In the visualized graphs, a node is one of the 15 primitives coded with markers shown at the bottom, where they are sorted by the frequency in the training set. For visualization purposes, a blue triangle marker differentiates a 1×1 convolution (equivalent to a fully-connected layer over channels) from other convolutions, but its primitive type is still just convolution. *Computed based on CIFAR-10.

| | IN-DISTRIBUTION | | OUT-OF-DISTRIBUTION | | | | |
|---|---|---|---|---|---|---|---|
| | TRAIN | VAL/TEST | WIDE | DEEP | DENSE | BN-FREE | RESNET/VIT |
| #graphs | $10^6$ | 500/500 | 100 | 100 | 100 | 100 | 1/1 |
| #cells | 4-18 | 4-18 | 4-18 | **10-36** | 4-18 | 4-18 | 16/12 |
| #channels | 16-128 | 32-128 | **128-1216** | 32-208 | 32-240 | 32-336 | 64/128 |
| #nodes ($|V|$) | 21-827 | 33-579 | 33-579 | **74-1017** | **57-993** | 33-503 | 161/114 |
| % w/o BN | 3.5% | 4.1% | 4.1% | 2.0% | 5.0% | **100%** | 0%/**100%** |
| #params(M)* | 0.01-3.1 | 2.5-35 | **39-101** | 2.5-15.3 | 2.5-8.8 | 2.5-7.7 | **23.5**/1.0 |
| avg degree | 2.3±0.1 | 2.3±0.1 | 2.3±0.1 | 2.3±0.1 | **2.4**±0.1 | **2.4**±0.1 | 2.2/2.3 |
| avg path | 14.5±4.8 | 14.5±4.9 | 14.7±4.9 | **26.2**±9.3 | 15.1±4.1 | 10.0±2.8 | 11.2/10.7 |

| marker primitive | conv | BN | sum | bias | group conv | concat | dilated gr. conv | LN | max pool | avg pool | MSA | SE | input | glob avg | pos enc |
|---|---|---|---|---|---|---|---|---|---|---|---|---|---|---|---|
| fraction in TRAIN (%) | 36.3 | 25.5 | 11.1 | 6.5 | 5.1 | 3.8 | 2.5 | 2.5 | 1.8 | 1.7 | 1.2 | 1.0 | 0.5 | 0.5 | 0.2 |

## 3   DEEPNETS-1M

The network design space of DARTS is limited by the number of unique operations that compose cells, and the low variety of stems and classification heads. Thus, many architectures are not realizable within this design space, including: VGG [25], ResNets [8], MobileNet [33] or more recent ones such as Visual Transformer (ViT) [20] and Normalization-free networks [36, 37]. Furthermore, DARTS does not define a procedure to sample random architectures. By addressing these two limitations we aim to expose our hypernetwork to diverse training architectures and permit its evaluation on common architectures, such as ResNet-50. We hypothesize that increased training diversity can improve hyper-networks' generalization to unseen architectures making it more competitive to iterative optimizers.

**Extending the network design space.** We extend the set of possible operations with non-separable 2D convolutions[3], Squeeze&Excite[4] (SE) [40] and Transformer-based operations [41, 20]: multihead self-attention (MSA), positional encoding and layer norm (LN) [42]. Each node (operation) in our graphs has two attributes: *primitive type* (e.g. convolution) and *shape* (e.g. 3×3×512×512). Overall, our extended set consists of 15 primitive types (Table 1). We also extend the diversity of the generated architectures by introducing VGG-style classification heads and ViT stems. Finally, to further increase architectural diversity, we allow the operations to not include batch norm (BN) [7] and permit networks without channel width expansion (e.g. as in [20]).

**Architecture generation process.** We generate different subsets of architectures (see the description of each subset in the next two paragraphs and in Table 1). For each subset depending on its purpose, we predefine a range of possible model depths (number of cells), widths and number of nodes per cell. Then, we sample a stem, a normal and reduction cell and a classification head. The internal structure of the normal and reduction cells is defined by uniformly sampling from all available operations. Due to a diverse design space it is extremely unlikely to sample the same architecture multiple times, but we ran a sanity check using the Hungarian algorithm [43] to confirm that (see Figure 6 in § A.2 for details).

**In-distribution (ID) architectures.** We generate a training set of $|\mathcal{F}| = 10^6$ architectures and validation/test sets of 500/500 architectures that follow the same generation rules and are considered to be ID samples. However, training on large architectures can be prohibitive, e.g. in terms of GPU memory. Thus, in the training set we allow the number of channels and, hence the total number of parameters, to be stochastically defined given computational resources. For example, to train

---

[3]Non-separable convolutions have weights of e.g. shape 3×3×512×512 as in ResNet-50. NAS works, such as DARTS and GHN, avoid such convolutions, since the separable ones [38] are more efficient. Non-separable convolutions are nevertheless common in practice and can often boost the downstream performance.

[4]The Squeeze&Excite operation is common in many efficient networks [39, 12].

our models we upper bound the number of parameters in the training architectures to around 3M by sampling fewer channels if necessary. In the evaluation sets, the number of channels is fixed. Therefore, this pre-processing step prior to training results in some distribution shift between the training and the validation/test sets. However, the shift is not imposed by our dataset.

**Out-of-distribution (OOD) architectures.** We generate five OOD test sets that follow different generation rules. In particular, we define WIDE and DEEP sets that are of interest due the stronger downstream performance of such nets in large-scale tasks [47, 48, 10]. These nets are often more challenging to train for fundamental [49, 50] or computational [51] reasons, so predicting their parameters might ease their subsequent optimization. We also define the DENSE set, since networks with many operations per cell and complex connectivity are underexplored in the literature despite their potential [27]. Next, we define the BN-FREE set that is of interest due to BN's potential negative side-effects [52, 53] and the difficulty or unnecessity of using it in some cases [54–56, 36, 37]. We finally add the RESNET/VIT set with two predefined image classification architectures: commonly-used ResNet-50 [8] and a smaller 12-layer version of the Visual Transformer (ViT) [20] that has recently received a lot of attention in the vision community. Please see § A.1 and § A.2 for further details and statistics of our DEEPNETS-1M dataset.

# 4 Improved Graph HyperNetworks: GHN-2

In this section, we introduce our three key improvements to the baseline GHN-1 described in § 2.2 (Fig. 1). These components are essential to predict stronger parameters on our task. For the empirical validation of the effectiveness of these components see ablation studies in § 5.1 and § C.2.1.

## 4.1 Differentiable Normalization of Predicted Parameters

When training the parameters of a given network from scratch using iterative optimization methods, the initialization of parameters is crucial. A common approach is to use He [57] or Glorot [58] initialization to stabilize the variance of activations across layers of the network. Chang et al. [59] showed that when the parameters of the network are instead predicted by a hypernetwork, the activations in the network tend to

Table 2: Parameter normalizations.

| Type of node $v$ | Normalization |
| --- | --- |
| Conv./fully-conn. | $\hat{\mathbf{w}}_p^v \sqrt{\beta/(C_{in}\mathcal{H}\mathcal{W})}$ |
| Norm. weights | $2 \times \text{sigmoid}(\hat{\mathbf{w}}_p^v/T)$ |
| Biases | $\tanh(\hat{\mathbf{w}}_p^v/T)$ |

explode or vanish. To address the issue of unstable network activations especially for the case of predicting parameters of diverse architectures, we apply *operation-dependent normalizations* (Table 2). We normalize convolutional and fully-connected weights by following the *fan-in* scheme of [57] (see the comparison to *fan-out* in § C.2.1): $\hat{\mathbf{w}}_p^v \sqrt{\beta/(C_{in}\mathcal{H}\mathcal{W})}$, where $C_{in}, \mathcal{H}, \mathcal{W}$ are the number of input channels and spatial dimensions of weights $\hat{\mathbf{w}}_p^v$, respectively; and $\beta$ is a nonlinearity specific constant following the analysis in [57]. The parameters of normalization layers such as BN and LN, as well as biases typically initialized with constants, are normalized by applying a squashing function with temperature $T$ to imitate the empirical distributions of models trained with SGD (see Table 2). These are differentiable normalizations, so that they are applied at training (and testing) time. Further analysis of our normalization and its stabilizing effect on activations is presented in § B.2.2.

## 4.2 Enhancing Long-range Message Propagation

Computational graphs often take the form of long chains (Table 1) with only a few incoming/outcoming edges per node. This structure might hinder long-range propagation of information between nodes [60]. Different approaches to alleviate the long-range propagation problem exist [61–63], including stacking GHNs in [24]. Instead we adopt simple graph-based heuristics in line with recent works [64, 65]. In particular, we add *virtual edges* between two nodes $v$ and $u$ and weight them based on the shortest

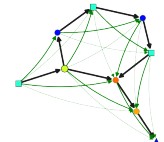

Figure 2: Virtual edges (in green) allow for better capture of global context.

path $s_{vu}$ between them (Fig. 2). To avoid interference with the *real* edges in the computational graph, we introduce a separate MLP$_{sp}$ to transform the features of the nodes connected through these virtual edges, and redefine the message passing of Equation 3 as:

$$\mathbf{m}_v^t = \sum\nolimits_{u \in \mathcal{N}_v^\pi} \text{MLP}(\mathbf{h}_u^t) + \sum\nolimits_{u \in \mathcal{N}_v^{(sp)}} \frac{1}{s_{vu}} \text{MLP}_{sp}(\mathbf{h}_u^t), \qquad (4)$$

where $\mathcal{N}_v^{(sp)}$ are neighbors satisfying $1 < s_{vu} \le s^{(max)}$, and $s^{(max)}$ is a hyperparameter. To maintain the same number of trainable parameters as in GHN-1, we decrease MLPs' sizes appropriately. Despite its simplicity, this approach is effective (see the comparison to stacking GHNs in § C.2.1).

### 4.3 Meta-batching Architectures During Training

GHN-1 updates its parameters $\theta$ based on a single architecture sampled for each batch of images (Equation 2). In vanilla SGD training, larger batches of images often speed up convergence by reducing gradient noise and improve model's performance [66]. Therefore, we define a meta-batch $b_m$ as the number of architectures sampled per batch of images. Both the parameter prediction and the forward/backward passes through the architectures in a meta-batch can be done in parallel. We then average the gradients across $b_m$ to update the parameters $\theta$ of $H_{\mathcal{D}}$: $\nabla_\theta \mathcal{L} = 1/b_m \sum_{i=1}^{b_m} \nabla_\theta \mathcal{L}_i$. Further analysis of the meta-batching effect on the training loss and convergence speed is presented in § B.2.3.

## 5 Experiments

We focus the evaluation of GHN-2 on our parameter prediction task (§ 5.1). In addition, we show beneficial side-effects of i) learning a stronger neural architecture representation using GHN-2 in analyzing networks (§ 5.2) and ii) predicting parameters for fine-tuning (§ 5.3). We provide further experimental and implementation details, as well as more results supporting our arguments in § C.

**Datasets.** We use the DEEPNETS-1M dataset of architectures (§ 3) as well as two image classification datasets $\mathcal{D}_1$ (CIFAR-10 [15]) and $\mathcal{D}_2$ (ImageNet [1]). CIFAR-10 consists of 50k training and 10k test images of size $32\times32\times3$ and 10 object categories. ImageNet is a larger scale dataset with 1.28M training and 50k test images of variable size and 1000 fine-grained object categories. We resize ImageNet images to $224\times224\times3$ following [19, 24]. We use 5k/50k training images as a validation set in CIFAR-10/ImageNet and 500 validation architectures of DEEPNETS-1M for hyperparameter tuning.

**Baselines.** Our baselines include GHN-1 and a simple MLP that only has access to operations, but not to the connections between them. This MLP baseline is obtained by replacing the GatedGNN with an MLP in our GHN-2. Since GHNs were originally introduced for small architectures of $\sim 50$ nodes and only trained on CIFAR-10, we reimplement[5] them and scale them up by introducing minor modifications to their decoder that enable their training on ImageNet and on larger architectures of up to 1000 nodes (see § B.1 for details). We use the same hyperparameters to train the baselines and GHN-2.

**Iterative optimizers.** In the parameter prediction experiments, we also compare our model to standard optimization methods: SGD and Adam [3]. We use off-the-shelf hyperparameters common in the literature [24, 19, 32, 67–69]. On CIFAR-10, we train evaluation architectures with SGD/Adam, initial learning rate $\eta = 0.025$ / $\eta = 0.001$, batch size $b = 96$ and up to 50 epochs. With Adam, we train only 300 evaluation architectures as a rough estimation of an average performance. On ImageNet, we train them with SGD, $\eta = 0.1$ and $b = 128$, and, for computational reasons (given 1402 evaluation architectures in total), we limit training with SGD to 1 epoch. We have also considered meta-optimizers, such as [17, 18]. However, we were unable to scale them to diverse and large architectures of our DEEPNETS-1M, since their LSTM requires a separate hidden state for every trainable parameter in the architecture. The scalable variants exist [70, 71], but are hard to reproduce without open source code.

**Additional experimental details.** We follow [24] and train GHNs with Adam, $\eta = 0.001$ and batch size of 64 images for CIFAR-10 and 256 for ImageNet. We train for up to 300 epochs, except for one experiment in the ablation studies, where we train one GHN with $b_m = 1$ eight times longer, i.e. for 2400 epochs. All GHNs in our experiments use $T = 1$ propagation (Equation 3), as we found the original $T = 5$ of [24] to be inefficient and it did not improve the accuracies in our task. GHN-2 uses $s^{(\mathrm{max})} = 50$ and $b_m = 8$ and additionally uses LN that slightly further improves results (see these ablations in § C.2.1). Model selection is performed on the validation sets, but the results in our paper are reported on the test sets to enable their direct comparison.

### 5.1 Parameter Prediction

**Experimental setup.** We trained our GHN-2 and baselines on the training architectures and training images, i.e. a separate model is trained for CIFAR-10 and ImageNet. According to our DEEPNETS-1M benchmark, we assess whether these models can generalize to unseen in-distribution (ID) and out-of-distribution (OOD) test architectures from our DEEPNETS-1M. We measure this generalization by predicting parameters for the test architectures and computing their classification accuracies on the test images of CIFAR-10 (Table 3) and ImageNet (Table 4). The evaluation architectures with batch norm (BN) have running statistics, which are not learned by gradient descent [7], and

---

[5]While source code for GHNs [24] is unavailable, we appreciate the authors' help in implementing some steps.

hence are not predicted by our GHNs. To alleviate that, we follow [24] and evaluate the networks with BN by computing per batch statistics with batch size of 64 images. This is further discussed in § C.1.

**Results.** Despite GHN-2 never observed the test architectures, GHN-2 predicts good parameters for them making the test networks perform surprisingly well on both image datasets (Tables 3 and 4). Our results are especially strong on CIFAR-10, where some architectures with predicted parameters achieve up to 77.1%, while the best accuracy of training with SGD for 50 epochs is around 15% more. We even show good results on ImageNet, where for some architectures we achieve a top-5 accuracy of up to 48.3%. While these results are low for direct downstream applications, they are remarkable for three main reasons. First, to train GHNs by optimizing Equation 2, we do not rely on the prohibitively expensive procedure of training the architectures $\mathcal{F}$ by SGD. Second, GHNs rely on a single forward pass to predict all parameters. Third, these results are obtained for unseen architectures, including the OOD ones. Even in the case of severe distribution shifts (e.g. ResNet-50[6]) and underrepresented networks (e.g. ViT[7]), our model still predicts parameters that perform better than random ones. On CIFAR-10, generalization of GHN-2 is particularly strong with a 58.6% accuracy on ResNet-50.

On both image datasets, our GHN-2 significantly outperforms GHN-1 on all test subsets of DEEPNETS-1M with more than a 20% absolute gain in certain cases, e.g. 36.8% vs 13.7% on the BN-FREE networks (Table 3). Exploiting the structure of computational graphs is a critical property of GHNs with the accuracy dropping from 66.9% to 42.2% on ID (and even more on OOD) architectures when we replace the GatedGNN of GHN-2 with an MLP. Compared to iterative optimization methods, GHN-2 predicts parameters achieving an accuracy similar to ~2500 and ~5000 iterations of SGD on CIFAR-10 and ImageNet respectively. In contrast, GHN-1 performs similarly to only ~500 and ~2000 (not shown in Table 4) iterations respectively. Comparing SGD to Adam, the latter performs worse in general except for the ViT architectures similar to [72, 20].

To report speeds on ImageNet in Table 4, we use a dedicated machine with a single NVIDIA V100-32GB and Intel Xeon CPU E5-1620 v4@ 3.50GHz. So for SGD these numbers can be reduced by using faster computing infrastructure and more optimal hyperparameters [73]. Using our setup,

Table 3: CIFAR-10 results of predicted parameters for unseen ID and OOD architectures of DEEPNETS-1M. Mean ($\pm$standard error of the mean) accuracies are reported (random chance $\approx$10%). [†]The number of parameter updates.

| METHOD | #upd[†] | ID-TEST | | OOD-TEST | | | | |
|---|---|---|---|---|---|---|---|---|
| | | avg | max | WIDE | DEEP | DENSE | BN-FREE | RESNET/VIT |
| MLP | 1 | 42.2±0.6 | 60.2 | 22.3±0.9 | 37.9±1.2 | 44.8±1.1 | 23.9±0.7 | 17.7/10.0 |
| GHN-1 | 1 | 51.4±0.4 | 59.9 | 43.1±1.7 | 48.3±0.8 | 51.8±0.9 | 13.7±0.3 | 19.2/**18.2** |
| GHN-2 | 1 | **66.9**±0.3 | **77.1** | **64.0**±1.1 | **60.5**±1.2 | **65.8**±0.7 | **36.8**±1.5 | **58.6**/11.4 |
| *Iterative optimizers (all architectures are ID in this case)* | | | | | | | | |
| SGD (1 epoch) | $0.5\times10^3$ | 46.1±0.4 | 66.5 | 47.2±1.1 | 34.2±1.1 | 45.3±0.7 | 18.0±1.1 | 61.8/34.5 |
| SGD (5 epochs) | $2.5\times10^3$ | 69.2±0.4 | 82.4 | 71.2±0.3 | 56.7±1.6 | 67.8±0.9 | 29.0±2.0 | 78.2/52.5 |
| SGD (50 epochs) | $25\times10^3$ | 88.5±0.3 | 93.1 | 88.9±1.2 | 84.5±1.2 | 87.3±0.8 | 45.6±3.6 | 93.5/75.7 |
| Adam (50 epochs) | $25\times10^3$ | 84.0±0.8 | 89.5 | 82.0±1.6 | 76.2±2.6 | 84.8±0.4 | 38.8±4.8 | 91.5/79.4 |

Table 4: ImageNet results on DEEPNETS-1M. Mean ($\pm$standard error of the mean) top-5 accuracies are reported (random chance $\approx$0.5%). [*]Estimated on ResNet-50 with batch size 128.

| METHOD | #upd | GPU sec. avg | CPU sec. avg | ID-TEST | | OOD-TEST | | | | |
|---|---|---|---|---|---|---|---|---|---|---|
| | | | | avg | max | WIDE | DEEP | DENSE | BN-FREE | RESNET/VIT |
| GHN-1 | 1 | 0.3 | 0.5 | 17.2±0.4 | 32.1 | 15.8±0.9 | 15.9±0.8 | 15.1±0.7 | 0.5±0.0 | **6.9**/0.9 |
| GHN-2 | 1 | 0.3 | 0.7 | **27.2**±0.6 | **48.3** | **19.4**±1.4 | **24.7**±1.4 | **26.4**±1.2 | **7.2**±0.6 | 5.3/**4.4** |
| *Iterative optimizers (all architectures are ID in this case)* | | | | | | | | | | |
| SGD (1 step) | 1 | 0.4 | 6.0 | 0.5±0.0 | 0.7 | 0.5±0.0 | 0.5±0.0 | 0.5±0.0 | 0.5±0.0 | 0.5/0.5 |
| SGD (5000 steps) | 5k | $2\times10^3$ | $3\times10^4$ | 25.6±0.3 | 50.7 | 26.2±1.4 | 13.2±1.1 | 25.4±1.1 | 4.8±0.8 | 34.8/24.3 |
| SGD (10000 steps) | 10k | $4\times10^3$ | $6\times10^4$ | 37.7±0.6 | 62.0 | 38.7±1.6 | 22.1±1.4 | 36.3±1.2 | 8.0±1.2 | 49.0/33.4 |
| SGD (100 epochs) | 1000k | $6\times10^{5*}$ | $6\times10^{7*}$ | — | | — | — | — | — | 92.9/72.2 |

---

[6]Large architectures with bottleneck layers such as ResNet-50 do not appear during training.

[7]Architectures such as ViT do not include BN and, except for the first layer, convolutions – the two most frequent operations in the training set.

SGD requires on average $10^4\times$ more time on a GPU ($10^5\times$ on a CPU) to obtain parameters that yield performance similar to GHN-2. As a concrete example, AlexNet [74] requires around 50 GPU hours (on our setup) to achieve a 81.8% top-5 accuracy, while on some architectures we achieve $\geq$48.0% in just 0.3 GPU seconds.

Table 5: Ablating GHN-2 on CIFAR-10. An average rank of the model is computed across all ID and OOD test architectures.

| MODEL | ID-TEST | OOD-TEST | AVG. RANK |
|---|---|---|---|
| GHN-2 | **66.9**±0.3 | **56.8**±0.8 | **1.9** |
| 1000 training architectures | 65.1±0.5 | 52.5±1.0 | 2.6 |
| No normalization (§ 4.1) | 62.6±0.6 | 47.1±1.2 | 3.9 |
| No virtual edges (§ 4.2) | 61.5±0.4 | 53.9±0.6 | 4.1 |
| No meta-batch ($b_m = 1$, § 4.3) | 54.3±0.3 | 47.5±0.6 | 5.5 |
| $b_m = 1$, train $8\times$ longer | 62.4±0.5 | 51.9±1.0 | 3.7 |
| No GatedGNN (MLP) | 42.2±0.6 | 32.2±0.7 | 7.4 |
| GHN-1 | 51.4±0.4 | 39.2±0.9 | 6.8 |

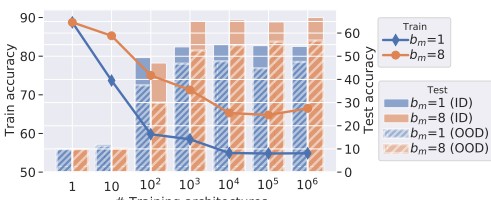

Figure 3: GHN-2 with meta batch $b_m = 8$ versus $b_m = 1$ for different numbers of training architectures on CIFAR-10.

Ablations (Table 5) show that all three components proposed in § 4 are important. Normalization is particularly important for OOD generalization with the largest drops on the WIDE and BN-FREE networks (see § C.2.1). Using meta-batching ($b_m = 8$) is also essential and helps stabilize training and accelerate convergence (see § B.2). We also confirm that the performance gap between $b_m = 1$ and $b_m = 8$ is not primarily due to the observation of more architectures, since the ablated GHN-2 with $b_m = 1$ trained eight times longer is still inferior. The gap between $b_m = 8$ and $b_m = 1$ becomes pronounced with *at least* 1k training architectures (Fig. 3). When training with fewer architectures (e.g. 100), the GHN with meta-batching starts to overfit to the training architectures. Given our challenging setup with unseen evaluation architectures, it is surprising that using 1k training architectures already gives strong results. However, OOD generalization degrades in this case compared to using all 1M architectures, especially on the BN-FREE networks (see § B.2). When training GHNs on just a few architectures, the training accuracy soars to the level of training them with SGD. With more architectures, it generally decreases indicating classic overfitting and underfitting cases.

## 5.2 Property Prediction

Representing computational graphs of neural architectures is a challenging problem [75–79]. We verify if GHNs are capable of doing that out-of-the-box in the property prediction experiments. We also experiment with architecture comparison in § C.2.4. Our hypothesis is that by better solving our parameter prediction task, GHNs should also better solve graph representation tasks.

**Experimental setup.** We predict the properties of architectures given their graph embeddings obtained by averaging node features[8]. We consider four such properties (see § C.2.3 for details):

- Accuracy on the "clean" (original) validation set of images;
- Accuracy on a corrupted set (obtained by adding the Gaussian noise to images following [53]);
- Inference speed (latency or GPU seconds per a batch of images);
- Convergence speed (the number of SGD iterations to achieve a certain training accuracy).

Estimating these properties accurately can have direct practical benefits. Clean and corrupted accuracies can be used to search for the best performing architectures (e.g. for the NAS task); inference speed can be used to choose the fastest network, so by estimating these properties we can trade-off accurate, robust and fast networks [12]. Convergence speed can be used to find networks that are easier to optimize. These properties correlate poorly with each other and between CIFAR-10 and ImageNet (§ C.2.3), so they require the model to capture different regularities of graphs. While specialized methods to estimate some of these properties exist, often as a NAS task [80–82, 30, 75], our GHNs provide a generic representation that can be easily used for many such properties. For each property, we train a simple regression model using graph embeddings and ground truth property values. We use 500 validation architectures of DEEPNETS-1M for training the regression model and tuning its hyperparameters (see § C.2.3 for details). We then use 500 testing architectures of DEEPNETS-1M to measure Kendall's Tau rank correlation between the predicted and ground truth property values similar to [80].

---

[8]A fixed size graph embedding for the architecture $a$ can be computed by averaging the output node features: $\mathbf{h}_a = \frac{1}{|V|}\sum_{v\in V}\mathbf{h}_v^T$, where $\mathbf{h}_a \in \mathbb{R}^d$ and $d$ is the dimensionality of node features.

**Additional baseline.** We compare to the Neural Predictor (NeuPred) [80]. NeuPred is based on directed graph convolution and is developed for accuracy prediction achieving strong NAS results. We train a separate such NeuPred for each property from scratch following their hyperparameters.

**Results.** GHN-2 consistently outperforms the GHN-1 and MLP baselines as well as NeuPred (Fig. 4). In § C.2.3, we also provide results verifying if higher correlations translate to downstream gains. For example, on CIFAR-10 by choosing the most accurate architecture according to the regression model and training it from scratch following [19, 24], we obtained a 97.26%($\pm$0.09) accuracy, which is competitive with leading NAS approaches, e.g. [19, 24, 32, 67–69]. In contrast, the

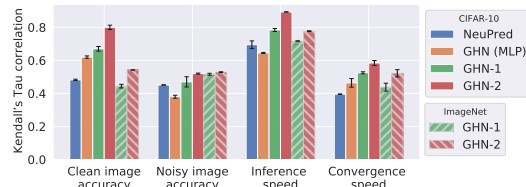

Figure 4: Property prediction of neural networks in terms of correlation (higher is better). Error bars denote the standard deviation across 5 runs.

network chosen by the regression model trained on the GHN-1 embeddings achieves 95.90%($\pm$0.08).

## 5.3 Fine-tuning Predicted Parameters

Neural networks trained on ImageNet and other large datasets have proven useful in diverse visual tasks in the transfer learning setup [83–87, 20]. Therefore, we explore how predicting parameters on ImageNet with GHNs compares to pretraining them on ImageNet with SGD in such a setup. We consider low-data tasks as they often benefit more from transfer learning [86, 87].

**Experimental setup.** We perform two transfer-learning experiments. The first experiment is fine-tuning the predicted parameters on 1,000 training samples (100 labels per class) of CIFAR-10. We fine-tune ResNet-50, Visual Transformer (ViT) and a 14-cell architecture based on the DARTS best cell [19]. The hyperparameters of fine-tuning (initial learning rate and weight decay) are tuned on 200 validation samples held-out of the 1,000 training samples. The number of epochs is fixed to 50 as in § 5.1 for simplicity. In the second experiment, we fine-tune the predicted parameters on the object detection task. We closely follow the experimental protocol and hyperparameters from [88] and train the networks on the Penn-Fudan dataset [89]. The dataset contains only 170 images and the task is to detect pedestrians. Therefore this task is also well suited for transfer learning. Following [88], we replace the backbone of a Faster R-CNN with one of the three architectures. To perform transfer learning with GHNs, in both experiments we predict the parameters of a given architecture using GHNs trained on ImageNet. We then replace the ImageNet classification layer with the target task-specific layers and fine-tune the entire network on the target task. We compare the results of GHNs to He's initialization [57] and the initialization based on pretraining the parameters on ImageNet with SGD.

Table 6: CIFAR-10 test set accuracies and Penn-Fudan object detection average precision (at IoU=0.50) after fine-tuning the networks using SGD initialized with different methods. Average results and standard deviations for 3 runs with different random seeds are shown. For each architecture, similar GHN-2-based and ImageNet-based results are bolded.[*]Estimated on ResNet-50.

| INITIALIZATION METHOD | GPU sec. to init.[*] | 100-SHOT CIFAR-10 | | | PENN-FUDAN OBJECT DETECTION | | |
|---|---|---|---|---|---|---|---|
| | | RESNET-50 | VIT | DARTS | RESNET-50 | VIT | DARTS |
| He's [57] | 0.003 | 41.0$\pm$0.4 | 33.2$\pm$0.3 | 45.4$\pm$0.4 | 0.197$\pm$0.042 | 0.144$\pm$0.010 | 0.486$\pm$0.035 |
| GHN-1 (trained on ImageNet) | 0.6 | 46.6$\pm$0.0 | 23.3$\pm$0.1 | 49.2$\pm$0.1 | 0.433$\pm$0.013 | 0.0$\pm$0.0 | 0.468$\pm$0.024 |
| GHN-2 (trained on ImageNet) | 0.7 | **56.4**$\pm$0.1 | **41.4**$\pm$0.6 | **60.7**$\pm$0.3 | **0.560**$\pm$0.019 | **0.436**$\pm$0.032 | **0.785**$\pm$0.032 |
| ImageNet (1k pretraining steps) | $6\times10^2$ | 45.4$\pm$0.3 | **44.3**$\pm$0.1 | 62.4$\pm$0.3 | 0.302$\pm$0.022 | 0.182$\pm$0.046 | **0.814**$\pm$0.033 |
| ImageNet (2.5k pretraining steps) | $1.5\times10^3$ | **55.4**$\pm$0.2 | 50.4$\pm$0.3 | 70.4$\pm$0.2 | **0.571**$\pm$0.056 | 0.322$\pm$0.073 | 0.823$\pm$0.022 |
| ImageNet (5 pretraining epochs) | $3\times10^4$ | 84.6$\pm$0.2 | 70.2$\pm$0.5 | 83.9$\pm$0.1 | 0.723$\pm$0.045 | 0.391$\pm$0.024 | 0.827$\pm$0.053 |
| ImageNet (final epoch) | $6\times10^5$ | 89.2$\pm$0.2 | 74.5$\pm$0.2 | 85.6$\pm$0.2 | 0.876$\pm$0.011 | **0.468**$\pm$0.023 | 0.881$\pm$0.023 |

**Results.** The CIFAR-10 image classification results of fine-tuning the parameters predicted by our GHN-2 are $\geq$10 percentage points better (in absolute terms) than fine-tuning the parameters predicted by GHN-1 or training the parameters initialized using He's method (Table 6). Similarly, the object detection results of GHN-2-based initialization are consistently better than both GHN-1 and He's initializations. The GHN-2 results are a factor of 1.5-3 improvement over He's for all the three architectures. Overall, the two experiments clearly demonstrate the practical value of predicting parameters using our GHN-2. Using GHN-1 for initialization provides relatively small gains or hurts convergence (for ViT). Compared to pretraining on ImageNet with SGD, initialization using GHN-2 leads to performance sim-

ilar to 1k-2.5k steps of pretraining on ImageNet depending on the architecture in the case of CIFAR-10. In the case of Penn-Fudan, GHN-2's performance is similar to $\geq$1k steps of pretraining with SGD. In both experiments, pretraining on ImageNet for just 5 epochs provides strong transfer learning performance and the final ImageNet checkpoints are only slightly better, which aligns with previous works [85]. Therefore, further improvements in the parameter prediction models appear promising.

# 6   Related Work

Our proposed parameter prediction task, objective in Equation 2 and improved GHN are related to a wide range of machine learning frameworks, in particular meta-learning and neural architecture search (NAS). Meta-learning is a general framework [16, 90] that includes meta-optimizers and meta-models, among others. Related NAS works include differentiable [19] and one-shot methods [12]. See additional related work in § D.

**Meta-optimizers.** Meta-optimizers [17, 18, 71, 91, 92] define a problem similar to our task, but where $H_{\mathcal{D}}$ is an RNN-based model predicting the gradients $\nabla \mathbf{w}$, mimicking the behavior of iterative optimizers. Therefore, the objective of meta-optimizers may be phrased as *learning to optimize* as opposed to our *learning to predict* parameters. Such meta-optimizers can have their own hyperparameters that need to be tuned for a given architecture $a$ and need to be run expensively (on the GPU) for many iterations following Equation 1.

**Meta-models.** Meta-models include methods based on MAML [93], ProtoNets [94] and auxiliary nets predicting task-specific parameters [95–98]. These methods are tied to a particular architecture and need to be trained from scratch if it is changed. Several recent methods attempt to relax the choice of architecture in meta-learning. T-NAS [99] combines MAML with DARTS [19] to learn both the optimal architecture and its parameters for a given task. However, the best network, $a$, needs to be trained using MAML from scratch. Meta-NAS [100] takes a step further and only requires fine-tuning of $a$ on a given task. However, the $a$ is obtained from a single meta-architecture and so its choice is limited, preventing parameter prediction for arbitrary $a$. CATCH [101] follows a similar idea, but uses reinforcement learning to quickly search for the best $a$ on the specific task. Overall meta-learning mainly aims at generalization *across tasks*, often motivated by the few-shot learning problem. In contrast, our parameter prediction problem assumes a single task (here an image dataset), but aims at generalization *across architectures* $a$ with the ability to predict parameters in a single forward pass.

**One-shot NAS.** One-shot NAS aims to learn a single "supernet" [102, 12, 103] that can be used to estimate the performance of smaller nets (subnets) obtained by some kind of pruning the supernet, followed by training the best chosen $a$ from scratch with SGD. Recent models, in particular BigNAS [12] and OnceForAll (OFA) [102], eliminate the need to train subnets. However, the fundamental limitation of one-shot NAS is poor scaling with the number of possible computational operations [24]. This limits the diversity of architectures for which parameters can be obtained. For example, all subnets in OFA are based on MobileNet-v3 [33], which does not allow to solve our more general parameter prediction task. To mitigate this, SMASH [104] proposed to predict some of the parameters using hypernetworks [14] by encoding architectures as a 3D tensor. Graph HyperNetworks (GHNs) [24] further generalized this approach to "arbitrary" computational graphs (DAGs), which allowed them to improve NAS results. GHNs focused on obtaining reliable subnetwork rankings for NAS and did not aim to predict large-scale performant parameters. We show that the vanilla GHNs perform poorly on our parameter prediction task mainly due to the inappropriate scale of predicted parameters, lack of long-range interactions in the graphs, gradient noise and slow convergence when optimizing Equation 2. Conventionally to NAS, GHNs were also trained in a quite constrained architecture space [105]. We expand the architecture space adopting GHNs for a more general problem.

# 7   Conclusion

We propose a novel framework and benchmark to learn and evaluate neural parameter prediction models. Our model (GHN-2) is able to predict parameters for very diverse and large-scale architectures in a single forward pass in a fraction of a second. The networks with predicted parameters yield surprisingly high image classification accuracy given the extremely challenging nature of our parameter prediction task. However, the accuracy is still far from networks trained with handcrafted optimization methods. Bridging the gap is a promising future direction. As a beneficial side-effect, GHN-2 learns a strong representation of neural architectures as evidenced by our property prediction evaluation. Finally, parameters predicted using GHN-2 trained on ImageNet benefit transfer learning in the low-data regime. This motivates further research towards solving our task.

# Acknowledgments

BK is thankful to Facebook AI Research for funding the initial phase of this research during his internship and to NSERC and the Ontario Graduate Scholarship used to fund the other phases of this research. GWT and BK also acknowledge support from CIFAR and the Canada Foundation for Innovation. Resources used in preparing this research were provided, in part, by the Province of Ontario, the Government of Canada through CIFAR, and companies sponsoring the Vector Institute: `http://www.vectorinstitute.ai/#partners`. We are thankful to Magdalena Sobol for editorial help. We are thankful to the Vector AI Engineering team (Gerald Shen, Maria Koshkina and Deval Pandya) for code review. We are also thankful to the reviewers for their constructive feedback.

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
