# Appendix

## Table of Contents

---

## A   DEEPNETS-1M Details

### A.1   Generating DEEPNETS-1M using DARTS

In this section, we elaborate on our description in § 2.1 about how DARTS [19] defines networks. We also elaborate on our discussion in § 3 about how we modify the DARTS framework to generate our DEEPNETS-1M dataset of architectures and summarize these modifications here, in Table 7. We visualize examples of architectures defined using DARTS [19] and corresponding computational graphs obtained using our code (Fig. 5).

**Overall architecture structure.** At a high level, all our ID and OOD networks are composed of a stem, repeated normal and reduction cells, global average pooling and a classification head (Fig. 5, (a)). We optionally sample fully-connected layers between the global pooling and the last classification layer and/or replace global pooling with fully connected layers, e.g. as in VGG [25]. The stem in DARTS, and in other NAS works, is predefined and fixed for each image dataset. We uniformly sample either a CIFAR-10 style or ImageNet style stem, so that our network space is unified for both image datasets. To prevent extreme GPU memory consumption when using a non-ImageNet stem for ImageNet images, we additionally use a larger stride in the stem that does not affect the graph structure. **At test time**, however, we can predict parameters for networks without these constraints, but the performance of the predicted parameters might degrade accordingly. For example, we can successfully predict parameters for ResNet-50 (Fig. 5, (e)), which has 13 normal and 3 reduction cells placed after the 3rd, 7th and 13th cells. ResNet-50's cells (Fig. 5, (d)) are similar to those of ResNet-18 (Fig. 5, (b)), but have $1 \times 1$ bottleneck layers.

**Within and between cell connectivity.** Within each cell and between them, there is a certain pattern to create connections in DARTS (Fig. 5, (b,d)): each cell receives features from the two previous

cells, each summation node can only receive features from two nodes, the last concatenation node can receive features from an arbitrary number of nodes. But due to the presence of the Zero ('none') and Identity ('skip connection') operations, we can enable any connectivity. We represent operations as nodes[9] and drop redundant edges and nodes. For example, if the node performs the Zero operation, we remove the edges connected to that node. This can lead to a small fraction of disconnected graphs, which we remove from the training/testing sets. If the node performs the Identity operation, we remove the node, but keep corresponding edges. We also omit ReLU operations and other nonlinearities in a graph representation to avoid significantly enlarging graphs, since the position of nonlinearities w.r.t. other operations is generally consistent across architectures (e.g. all convolutions are preceded by ReLUs except the first layer). This leads to graphs visualized in Fig. 5, (c,e).

**Operations.** The initial choice of operations in DARTS is quite standard in NAS. In normal cells, each operation returns the tensor of the same shape as it receives. So any differentiable operation that can preserve the shape of the input tensor can be used, therefore extending the set of operations is relatively trivial. In reduction cells, spatial resolution is reduced by a factor of 2, so some operations can have stride 2 there. In both cells, there are summation and concatenation nodes that aggregate features from several operations into one tensor. Concatenation (across channels) is used as the final node in a cell. To preserve channel dimensions after concatenating many features, $1\times1$ convolutions are used as needed. For the Squeeze&Exicte (SE) and Visual Transformer (ViT) operations, we use open source implementations[10] with default configurations, e.g. 8 heads in the multihead self-attention of ViT.

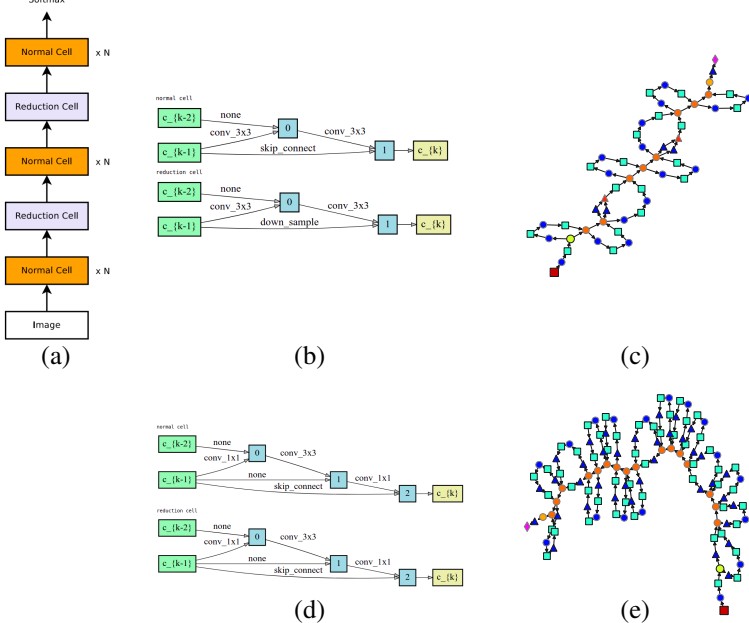

Figure 5: **(a)** Network's high-level structure introduced in [29] and employed by many following papers on network design, including DARTS [19] and ours, where N$\geq$ 1; **(b)** A residual block [8] in terms of DARTS normal and reduction cells, where green nodes denote outputs from the two previous cells, blue nodes denote summation, a yellow node denotes concatenation[†]; edges denote operations, 'none' indicates dropping the edge[‡]; the reduction cell has the same structure in ResNets, but decreases spatial resolution by 2 using a downsample operation and stride 2 in operations, at the same time, optionally increasing the channel dimensionality by 2. **(c)** The result of combining (a) and (b) for 8 cells using our code to build an analogous of the ResNet-18 architecture[⋆]. **(d)** A residual block of ResNet-50 with $1\times1$ bottleneck layers defined using DARTS and **(e)** the graph built using our code, where 3 reduction cells are placed as in the original ResNet-50 architecture.

[9]In DARTS, the operations are placed on the edges, except for inputs, summations and concatenations (Fig. 5).
[10]SE: https://github.com/ai-med/squeeze_and_excitation/blob/master/squeeze_and_excitation/squeeze_and_excitation.py, ViT: https://github.com/lucidrains/vit-pytorch/blob/main/vit_pytorch/vit.py

[†]Concatenation is redundant in ResNets and removed from our graphs due to only one input node in cells.

[‡]In ResNets [8], there is no skip connection between the input of a given cell and the output of the cell before the previous one.

[⋆]Note that ResNets of [8] commonly employed in practice have 3 reduction cells instead of 2 and have other minor differences (e.g. the order of convolution, BN and ReLU, down sampling convolution type, bottleneck layers, etc.). We still can predict parameters for them, but such architectures would be further away from the training distribution, so the predicted parameters might have significantly lower performance.

Table 7: Summary of differences between the DARTS design space and ours. *Means implementation-wise possibility of predicting parameters given a trained GHN, and does not mean our testing ID architectures, which follow the training design protocol. Overall, from the implementation point of view, our trained GHNs allow to predict parameters for arbitrary DAGs composed of our 15 primitives with the parameters of arbitrary shapes¶. We place ResNet-50 in a separate column even though it is one of the evaluation architectures of DEEPNETS-1M, because it has different properties as can be seen in the table.

| PROPERTY | DARTS | DEEPNETS-1M | RESNET-50 | TESTING GHN* |
|---|---|---|---|---|
| Unified style across image datasets | ✗ | ✓ | ✗ | ✓ |
| VGG style classification heads [25] | ✗ | ✓ | ✗ | ✓ |
| Visual Transformer stem [20] | ✗ | ✓ | ✗ | ✓ |
| Channel expansion ratio | 2 | 1 or 2 | 2 | arbitrary |
| Bottleneck layers (e.g. in ResNet-50 [8]) | ✗ | ✗ | ✓ | ✓ |
| Reduction cells position (w.r.t. total depth) | 1/3, 2/3 | 1/3, 2/3 | 3,7,17 cells | arbitrary |
| Networks w/o $1\times1$ preprocessing layers in cells | ✗ | ✓ | ✓ | ✓ |
| Networks w/o batch norm | ✗ | ✓ | ✗ | ✓ |

## A.2 DEEPNETS-1M Statistics

We show the statistics of the key properties of our DEEPNETS-1M in Fig. 6 and more examples of computational graphs for different subsets in Fig. 7.

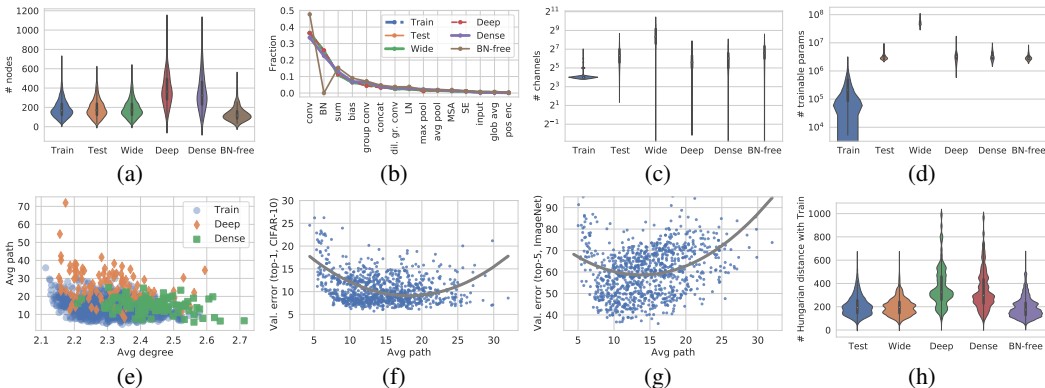

Figure 6: Visualized statistics of DEEPNETS-1M. (**a**) A violin plot of the number of nodes showing the distribution shift for the DEEP and DENSE subsets. (**b**) Node types (primitives) showing the distribution shift for the BN-FREE subset. (**c**) Number of initial channels in networks (for TRAIN, the number is for training GHNs on CIFAR-10). Here, the distribution shift is present for all test subsets (due to computational challenges of training GHNs on wide architectures), but the largest shift is for WIDE. (**d**) Total numbers of trainable parameters in case of CIFAR-10, where the distribution shifts are similar to those for the number of channels. (**e**) Average shortest path length versus average node degree (other subsets that are not shown follow the distribution of TRAIN), confirming that nodes of the DENSE subset have generally more dense connections (larger degrees), while in DEEP the networks are deeper (the shortest paths tend to be longer). (**f**) The validation error for 1000 VAL +TEST architectures (trained with SGD for 50 epochs) versus their average shortest path lengths, indicating the "sweet spot" of architectures with strong performance (same axes as in [46]). (**g**) Same as (f), but with the y axis being the top-5 validation error on ImageNet of the same architectures trained for 1 epoch according to our experiments. (**h**) The distribution of the distances between the architectures of a given test subset and the ones from a subset of TRAIN computed using the Hungarian algorithm [43], confirming that the evaluation architectures are different from the training ones.

---

¶While we predefine a wide range of possible shapes in our GHNs according to § B.1, in the rare case of using the shape that is not one of the predefined values, we use the closest values, which worked reasonably well in many cases.

| IN-DISTRIBUTION | OUT-OF-DISTRIBUTION | | | |
| TRAIN/VAL/TEST | WIDE | DEEP | DENSE | BN-FREE |

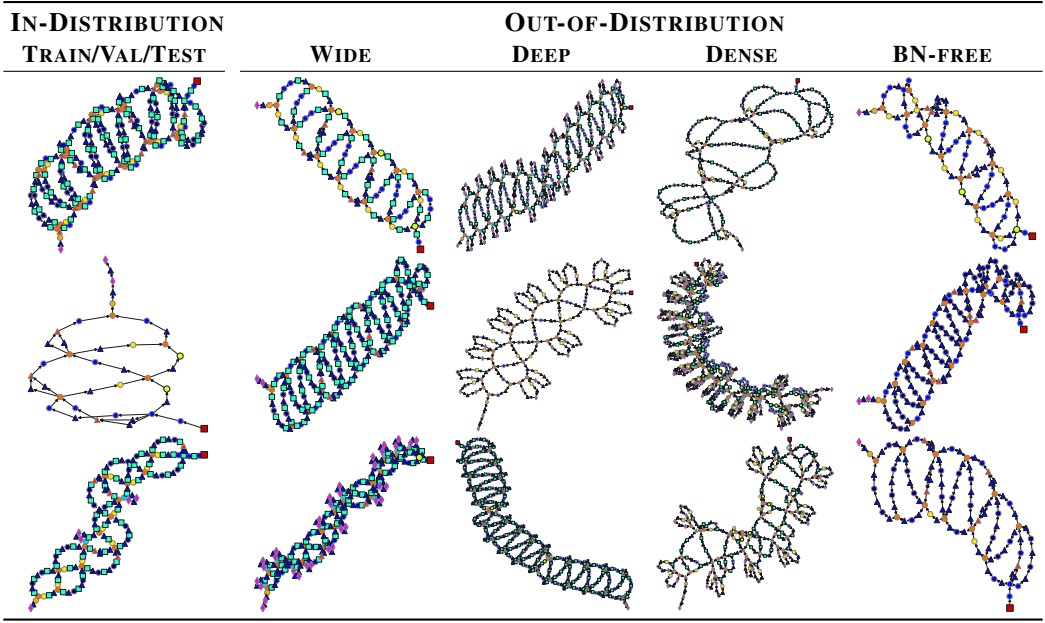

Figure 7: Examples of graphs in each subset of our DEEPNETS-1M visualized using NetworkX [44].

# B  GHN Details

## B.1  Baseline GHN: GHN-1

GHNs were designed for NAS, which typically make strong assumptions about the choice of operations and their possible dimensions to make search and learning feasible. For example, non-separable 2D convolutions (e.g. with weights like $512{\times}512{\times}3{\times}3$ in ResNet-50) are not supported. Our parameter prediction task is more general than NAS, and tackling it using the vanilla GHNs of [24] is not feasible (mainly, in terms of GPU memory and training efficiency) as we show in § C.2.1 (Table 8). So we first make the following modifications to GHNs and denote this baseline as GHN-1.

1. **Compact decoder:** We support the prediction of full 4D weights of shape <*out channels* $\times$ *input channels* $\times$ *height* $\times$ *width*> that is required for non-separable 2D convolutions. Using an MLP decoder of vanilla GHNs [24] would require it to have a prohibitive number of parameters (e.g. ∼4 billion parameters, see Table 8). To prevent that, we use an MLP decoder only to predict a small 3D tensor and then apply a $1{\times}1$ convolutional layer across the channels to increase their number followed by reshaping it to a 4D tensor.

2. **Diverse channel dimensions:** To enable prediction of parameters with channel dimensions larger than observed during training we implement a simple tiling strategy similar to [14]. In particular, instead of predicting a tensor of the maximum shape that has to be known prior training (as done in [24]), we predict a tensor with fewer channels, but tile across channel dimensions as needed. Combining this with #1 described above, our decoder first predicts a tensor of shape $128{\times}\mathcal{S}{\times}\mathcal{S}$, where $\mathcal{S} = 11$ for CIFAR-10 and $\mathcal{S} = 16$ for ImageNet. Then, the $1{\times}1$ convolutional decoder with 256 hidden and 4096 output units transforms this tensor to the tensor of shape $64{\times}64{\times}\mathcal{S}{\times}\mathcal{S}$. Modifications #1 and #2 can be viewed as strong regularizers that can hinder expressive power of a GHN and the parameters it predicts, but on the other side permit a more generic and efficient model with just around 1.6M-2M parameters.

3. **Fewer decoders:** One alternative strategy to reduce the number of parameters in the decoder of GHNs is to design multiple specialized decoders. However, this strategy does not scale well with adding new operations. Our modifications #1 and #2 allow us to have only three decoders: for convolutional and fully connected weights, for 1D weights and biases, such as affine transformations in normalization layers, and for the classification layer.

4. **Shape encoding:** The vanilla GHNs does not leverage the information about channel dimensionalities of parameters in operations. For example, the vanilla GHNs only differentiate between $3\times3$ and $5\times5$ convolutions, but not between $512\times512\times3\times3$ and $512\times256\times3\times3$. To alleviate that, we add two shape embedding layers: one for the spatial and one for the channel dimensions. For the spatial dimensions (height and width of convolutional weights) we predefine 11 possible values from 1 to $S$. For the channel dimensions (input and output numbers of channels) we predefine 392 possible values from 1 to 8192. We describe 3D, 2D and 1D tensors as special cases of 4D tensors using the value of 1 for missing dimensionalities (e.g. $10\times1\times1\times1$ for CIFAR-10 biases in the classification layer). The shape embedding layers transform the input shape into four (two for spatial and two for channel dimensions) 8-dimensional learnable vectors that are concatenated to obtain a 32-dimensional vector. This vector is summed with a 32-dimensional vector encoding one of the 15 operation types (primitives). In the rare case of feeding the shape that is not one of the predefined values, we look up for the closest value. This can work well in some cases, but can also hurt the quality of predicted parameters, so some continuous encoding as in [41] can be used in future work.

## B.2    Our improved GHN: GHN-2

### B.2.1    GHN-2 Architecture

Our improved GHN-2 is obtained by adding to GHN-1 the differentiable normalization of predicted parameters, virtual edges (and the associated 'mlp_ve' module, see the architecture below), meta-batching and layer normalization (and the associated 'ln' module). A high-level PyTorch-based overview of the GHN-2 model architecture used to predict the parameters for ImageNet is shown below (see our code for implementation details).

```
(ghn): GHN(  # our hypernetwork H_D with total 2,319,752 parameters (theta)
(gcn): GCNGated(  # Message Passing for Equations 3 and 4
(mlp): MLP(
(fc): Sequential(
(0): Linear(in_features=32, out_features=16, bias=True)
(1): ReLU()
(2): Linear(in_features=16, out_features=32, bias=True)
(3): ReLU()
)
)
(mlp_ve): MLP(  # only in GHN-2 (see Eq. 4)
(fc): Sequential(
(0): Linear(in_features=32, out_features=16, bias=True)
(1): ReLU()
(2): Linear(in_features=16, out_features=32, bias=True)
(3): ReLU()
)
)
(gru): GRUCell(32, 32)    # all GHNs use d=32 for input, hidden and output node feature dimensionalities
)
(ln): LayerNorm((32,), eps=1e-05, elementwise_affine=True)  # only in GHN-2
(shape_embed_spatial): Embedding(11, 8)  # encodes the spatial shape of predicted params
(shape_embed_channel): Embedding(392, 8)  # encodes the channel shape of predicted params
(op_embed): Embedding(16, 32)  # 15 primitives plus one extra embedding for dummy nodes
(decoder): Decoder(
(fc): Sequential(
(0): Linear(in_features=32, out_features=32768, bias=True)
(1): ReLU()
)  # predicts a 128x16x16 tensor
(conv): Sequential(
(0): Conv2d(128, 256, kernel_size=(1, 1), stride=(1, 1))
(1): ReLU()
(2): Conv2d(256, 4096, kernel_size=(1, 1), stride=(1, 1))
)  # predicts a 64x64x16x16 tensor
```

```
(class_layer_predictor): Sequential(
(0): ReLU()
(1): Conv2d(64, 1000, kernel_size=(1, 1), stride=(1, 1))
)  # predicts 64x1000 classification weights given the 64x64x16x16 tensor
)
(norm_layers_predictor): Sequential(
(0): Linear(in_features=32, out_features=64, bias=True)
(1): ReLU()
(2): Linear(in_features=64, out_features=128, bias=True)
)  # predicts 1x64 weights and 1x64 biases given the 1x32 node embeddings
(bias_predictor): Sequential(
(0): ReLU()
(1): Linear(in_features=64, out_features=1000, bias=True)
)  # predicts 1x1000 classification biases given the 64x64x16x16 tensor
)
```

### B.2.2 Differentiable Normalization of Predicted Parameters

We analyze in more detail the effect of normalizing predicted parameters (Fig. 8).

**Setup.** As we discussed in § 4.1, Chang et al. [59] proposed a method to initialize a hypernetwork to stabilize the activations in the network for which the parameters are predicted. However, this technique requires knowing upfront the shapes of the predicted parameters, and therefore is not applicable out-of-the-box in our setting, where we predict the parameters of diverse architectures with arbitrary shapes of weights. So, instead we apply operation-dependent normalizations. We analyze the effect of this normalization by taking a GHN in the beginning and end of training on CIFAR-10 and predicting parameters of ResNet-50. To compute the variance of activations in the ResNet-50, we forward pass a batch of test images through the predicted parameters.

**Observations.** The activations obtained using GHN-1 explode after training, which aligns with the analysis of [59] (this is more obvious on the left of Fig. 8, where a linear scale is used on the y axis). For GHN-2, in the beginning of training the activations match the ones of ResNet-50 initialized randomly using Kaiming He's method [57], which validates the choice of our normalization equations in § 4.1. By the end of training, the activations of models for the random-based and the GHN-2-based cases decrease (perhaps, due to the weight decay), however, the ones of GHN-2 reduce less, indicating that the predicted parameters do not reach the state of those trained with SGD from scratch. In contrast, the activations corresponding to GHN-1 have small values in the beginning, but explode by the end of training. Matching the activations of the models trained with SGD can be useful to improve training of GHNs and, for example, to make fine-tuning of predicated parameters easier as we show in § 5.3, where the parameters predicted by GHN-1 are shown difficult to be fine-tuned with SGD.

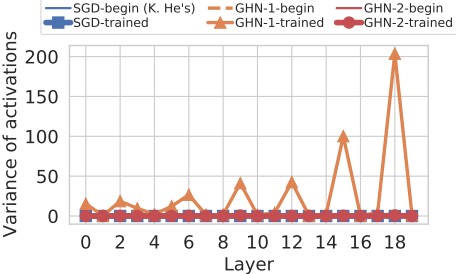 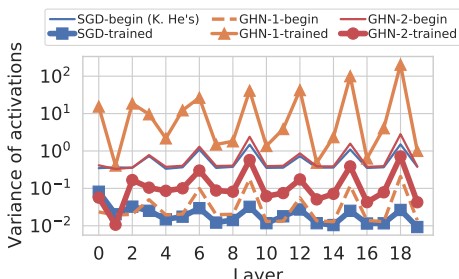

Figure 8: The effect of normalizing predicted parameters on the variance of activations in the first several layers of ResNet-50: a linear (**left**) and log (**right**) scale on the y axis.

### B.2.3 Meta-batching

We analyze how meta-batching affects the training loss when training GHNs on CIFAR-10 (Fig. 9). The loss of the GHN-2 with $b_m = 8$ is less noisy and is lower throughout the training compared to using $b_m = 1$. In fact, the loss of $b_m = 1$ is unstable to the extent that oftentimes the training fails

due to the numerical overflow, in which case we ignore the current architecture and resample a new one. For example, the standard deviation of gradient norms with $b_m = 8$ is significantly lower than with $b_m = 1$: 18 vs 145 with means 2.7 and 7.4 respectively. Training the model with $b_m = 1$ eight times longer (Fig. 9, right) boosts the performance of predicted parameters (Table 8), but still does not reach the level of $b_m = 8$; it also still suffers from the aforementioned numerical issues and does not leverage the parallelism of $b_m = 8$ (all architectures in a meta-batch can be processed in parallel). Further increasing the meta-batch size is an interesting avenue for future research.

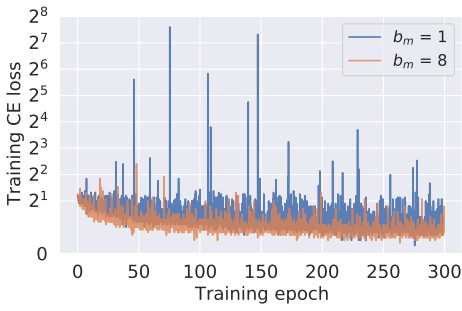 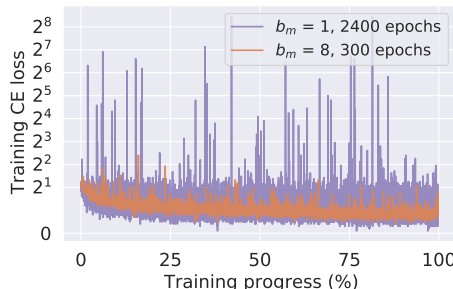

Figure 9: Effect of using more architectures per batch of images on the training loss (**left**) and comparison to training longer (**right**). In the figure on the right, we shrink the x axis of the $b_m = 1$ case, so that both plots can be compared w.r.t. the total number of epochs.

## C  Additional Experiments and Details

### C.1  Experimental Details

**Training GHNs.**   We train all GHNs for 300 epochs using Adam with an initial learning rate 0.001, batch size of 64 images for CIFAR-10 (256 for ImageNet), weight decay of 1e-5. The learning rate is decayed after 200 and 250 epochs. On ImageNet GHN-2's validation accuracy plateaued faster (in terms of training epochs), so we stopped training after 150 epochs decaying it after 100 and 140 epochs.

**Evaluation of networks with BN using GHNs.**   While our GHNs predict all *trainable* parameters, batch norm networks have running statistics that *are not learned by gradient descent* [7], and so are not predicted by GHNs. To obtain these statistics, we evaluate the networks with BN by computing per batch statistics on the test set as in [24] using a batch size 64. Another alternative we have considered is updating the running statistics by forward passing several batches of the training or testing images through each evaluation network (no labels, gradients or parameter updates are required for this stage). For example on CIFAR-10 if we forward pass 200 batches of the training images (takes less than 10 seconds on a GPU), we obtain a higher accuracy of 71.7% on ID-TEST compared to 66.9% when the former strategy is used. On ImageNet, the difference between the two strategies is less noticeable. For both strategies, the results can slightly change depending on the order of images for which the statistics is estimated.

### C.2  Additional Results

#### C.2.1  Additional Ablations

We present results on CIFAR-10 for different variants of GHNs (Table 8) in addition to those presented in Tables 3 and 5. Overall, different GHN variants show worse or comparable results on all evaluation architectures compared to our GHN-2, while in some cases having too many trainable parameters making training infeasible in terms of GPU memory or being less efficient to train (e.g. with $T = 5$ propagation steps). For ViT, GHN-2 is worse compared to other GHN variants, which requires further investigation.

Table 8: CIFAR-10 results of predicted parameters for the evaluation architectures of DEEPNETS-1M. Mean (±standard error of the mean) accuracies are reported. Different ablated GHN-2 models are evaluated. *GPU seconds per a batch of 64 images and $b_m$ architectures. **In [24] the GHNs have fewer parameters due to a more constrained network design space (as discussed in B.1) and applying specialized decoders for different operations. The best result in each column is bolded, the best results with $b_m = 1$ (excluding training $8 \times$ longer) are underlined.

| MODEL | Norm $\hat{w}_p$ | Virt. edges | M. batch $b_m = 8$ | #GHN params | Train. speed* | ID-TEST avg | ID-TEST max | OOD-TEST WIDE | DEEP | DENSE | BN-FREE | RESNET/VIT |
|---|---|---|---|---|---|---|---|---|---|---|---|---|
| GHN-2 | ✓ | ✓ | ✓ | 1.6M | 3.6 | **66.9**±0.3 | 77.1 | **64.0**±1.1 | **60.5**±1.2 | **65.8**±0.7 | 36.8±1.5 | **58.6**/11.4 |
| | | | | | | | | | | | | |
| 1000 archs | ✓ | ✓ | ✓ | 1.6M | 3.6 | 65.1±0.5 | **78.4** | 61.5±1.6 | 56.0±1.5 | 65.0±0.9 | 27.6±1.1 | 58.2/10.5 |
| 100 archs | ✓ | ✓ | ✓ | 1.6M | 3.6 | 47.1±0.8 | 77.1 | 38.8±1.9 | 28.3±1.6 | 41.9±1.5 | 11.0±0.2 | 38.7/10.3 |
| No normalization | ✗ | ✓ | ✓ | 1.6M | 3.6 | 62.6±0.6 | 75.9 | 52.3±2.1 | 59.5±1.1 | 62.3±1.2 | 14.4±0.4 | 58.3/17.0 |
| No virtual edges | ✓ | ✗ | ✓ | 1.6M | 3.6 | 61.5±0.4 | 73.2 | 58.2±1.0 | 55.0±0.9 | 61.5±0.6 | **40.8**±0.8 | 41.9/12.1 |
| No LayerNorm | ✓ | ✓ | ✓ | 1.6M | 3.6 | 64.5±0.4 | 75.9 | 62.4±1.1 | 59.0±1.1 | 64.6±0.6 | 39.6±1.2 | 55.1/8.9 |
| No GatedGNN (MLP) | ✓ | ✓ | ✓ | 1.6M | 1.5 | 42.2±0.6 | 60.2 | 22.3±0.9 | 37.9±1.2 | 44.8±1.1 | 23.9±0.7 | 17.7/10.0 |
| Train 8× longer | ✓ | ✓ | ✗ | 1.6M | 0.7 | 62.4±0.5 | 75.8 | 63.0±1.3 | 58.0±1.3 | 62.1±0.9 | 24.5±0.7 | 57.0/14.6 |
| | | | | | | | | | | | | |
| No Meta-batch ($b_m = 1$) | ✓ | ✓ | ✗ | 1.6M | 0.7 | 54.3±0.3 | 63.0 | 53.1±0.8 | 51.9±0.6 | 53.4±0.5 | 31.7±0.8 | 50.6/17.8 |
| No Shape Encoding | ✓ | ✓ | ✗ | 1.6M | 0.7 | 53.1±0.4 | 61.7 | 52.4±0.9 | 51.4±0.7 | 53.5±0.6 | 24.7±0.8 | 31.5/14.0 |
| No virtual edges (VE) | ✓ | ✗ | ✗ | 1.6M | 0.7 | 51.7±0.4 | 62.0 | 49.7±0.8 | 47.4±0.8 | 52.0±0.8 | 24.5±0.5 | 34.2/14.7 |
| +Stacked GHN, no VE | ✓ | ✗ | ✗ | 1.6M | 0.7 | 52.2±0.4 | 63.6 | 51.3±0.9 | 46.9±0.9 | 52.3±0.7 | 20.2±0.6 | 44.5/15.4 |
| +Stacked GHN | ✓ | ✓ | ✗ | 1.6M | 0.9 | 53.1±0.4 | 61.3 | 51.5±1.1 | 50.9±0.7 | 53.5±0.7 | 23.1±0.7 | 42.7/15.1 |
| $\pi$ = only fw (Eq. 3) | ✓ | ✓ | ✗ | 1.6M | 0.4 | 53.9±0.3 | 62.5 | 51.2±1.0 | 51.7±0.7 | 54.3±0.5 | 31.0±0.7 | 49.9/11.5 |
| $T = 5$ (Eq. 3) | ✓ | ✓ | ✗ | 1.6M | 2.6 | 54.4±0.4 | 63.3 | 52.8±1.0 | 50.4±0.9 | 53.4±0.8 | 22.6±1.0 | 50.1/10.1 |
| Fan-out | ✓ | ✓ | ✗ | 1.6M | 0.7 | 53.8±0.4 | 63.6 | 52.6±1.0 | 51.2±0.8 | 54.3±0.8 | 19.8±0.6 | 48.5/11.1 |
| MLP decoder | ✓ | ✓ | ✗ | 32M | 0.7 | 53.1±0.4 | 64.0 | 52.9±1.0 | 52.5±0.7 | 54.0±0.8 | 22.1±0.5 | 44.1/16.3 |
| No tiling | ✓ | ✓ | ✗ | 135M | | | | out of GPU memory | | | | |
| No tiling, MLP decoder (as in a vanilla GHN [24]) | ✓ | ✓ | ✗ | 4.1B** | | | | out of GPU memory | | | | |
| | | | | | | | | | | | | |
| GHN-1 | ✗ | ✗ | ✗ | 1.6M | 0.6 | 51.4±0.4 | 59.9 | 43.1±1.7 | 48.3±0.8 | 51.8±0.9 | 13.7±0.3 | 19.2/**18.2** |
| GHN-1 + LayerNorm | ✗ | ✗ | ✗ | 1.6M | 0.6 | 50.1±0.5 | 58.9 | 43.8±1.4 | 47.5±0.8 | 50.8±1.0 | 11.4±0.2 | 49.2/16.3 |

### C.2.2 Generalization Properties

On the OOD subsets, GHN results are lower than on ID-TEST as expected, so we inspect in more detail *how* performance changes with an increased distribution shift (Fig. 10). For example, training wider nets with SGD leads to similar or better performance, perhaps, due to increased capacity. However, GHNs degrade with larger width, since wide architectures are underrepresented during training for computational reasons (Table 1, Fig. 6). As for the depth and number of nodes, there is a certain range of values ("sweet spot") with higher performance. For architectures without batch norm (Fig. 10, the very right column), the results of GHN-2 are strong starting from a certain depth ($\geq$ 8-10 cells) matching the ones of training with SGD from scratch (for 50 epochs on CIFAR-10 and 1 epoch on ImageNet). This can be explained by the difficulty of training models without BN from scratch with SGD, while parameter prediction with GHN-2 is less affected by that. Generalization appears to be worse on ImageNet, perhaps due to its more challenging nature.

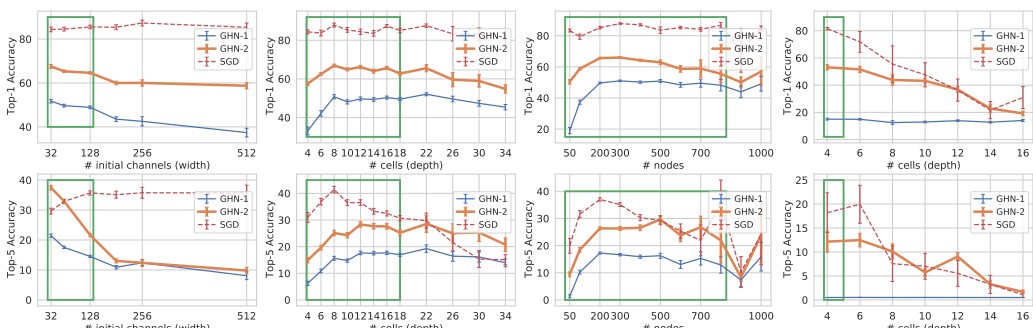

Figure 10: Generalization performance w.r.t. (from left to right): width, depth, number of nodes and depths for architectures without batch norm. A green rectangle denotes the training regime; bars are standard errors of the mean. Top row: CIFAR-10, bottom row: ImageNet.

### C.2.3   Property Prediction

In § 5.2, we experiment with four properties of neural architectures that can be estimated given an architecture and image dataset:

1. "Clean" classification accuracy measured on the validation sets of CIFAR-10 and ImageNet.

2. Classification accuracy on the corrupted images, which is created by adding the Gaussian noise of medium intensity to the validation images following [53][11]: with zero mean and the standard deviation of 0.08 on CIFAR-10 and 0.18 on ImageNet.

3. The inference speed is measured as GPU seconds required to process the entire validation set.

4. For the convergence speed, we measure the number of SGD iterations to achieve a training top-1 accuracy of 95% on CIFAR-10 and top-5 accuracy of 10% on ImageNet.

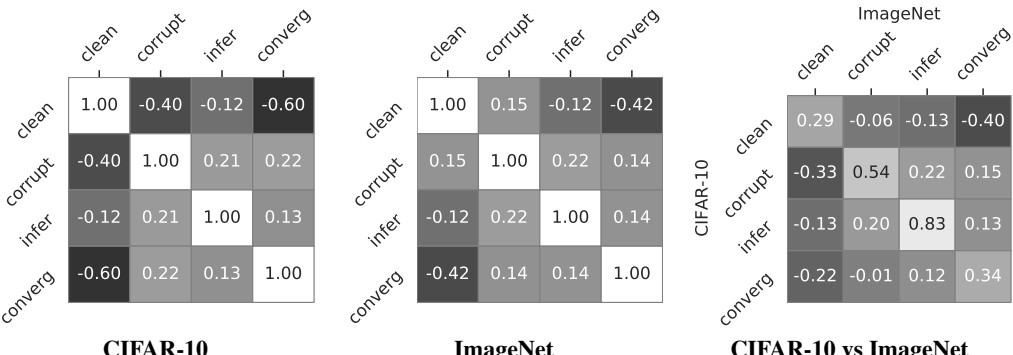

**CIFAR-10**          **ImageNet**          **CIFAR-10 vs ImageNet**

Figure 11: Cross correlation (Kendall's Tau) between ground truth values of properties.

**Ground truth property values.**   We first obtain ground truth values for each property for each of the 500 ID-VAL and 500 ID-TEST architectures of DEEPNETS-1M trained from scratch with SGD for 50 epochs (on CIFAR-10) and 1 epoch (on ImageNet) as described in § 5. The ground truth values between these properties and between CIFAR-10 and ImageNet correlate poorly (Fig. 11). Interestingly, on CIFAR-10 the architectures that rank higher on the clean images generally rank lower on the corrupted set and vice verse (correlation is -0.40). On ImageNet the correlation between these two properties is positive, but low (0.15). Also, the transferability of architectures between CIFAR-10 and ImageNet is poor contrary to a common belief, e.g. with the correlation of 0.29 on the clean and -0.06 on the corrupted sets. However, this might be due to training on ImageNet only for 1 epoch. The networks that converge faster have generally worse performance on the clean set, but tend to perform better on the corrupted set. The networks that classify images faster (have higher inference speed), also tend to perform better on the corrupted set. Investigating the underlying reasons for these relationships as well as extending the set of properties would be interesting in future work.

**Estimating properties by predicting parameters.**   A straightforward way to estimate this kind of properties using GHNs is by predicting the architecture parameters and forward passing images as was done in [24] for accuracy. However, this strategy has two issues: (a) the performance of parameters predicted by GHNs is strongly affected by the training distribution of architectures (Fig. 10); (b) estimating properties of thousands of networks for large datasets such as ImageNet can be time consuming. Regarding (a), for example the rank correlation on CIFAR-10 between the accuracy of the parameters predicted by GHN-2 and those trained with SGD is only 0.4 (down from 0.8 obtained with the regression model, Fig. 4).

**Training regression models.**   To report the results in Fig. 4, we treat the 500 ID-VAL architectures of DEEPNETS-1M as the training ones in this experiment. We train a simple regression model for each property using graph embeddings (obtained using MLP, GHN-1 or GHN-2) and ground truth property values of these architectures. We use Support Vector Regression[12] with the RBF kernel and tune hyperparameters (C, gamma, epsilon) on these 500 architectures using 5-fold cross-validation.

---

[11]https://github.com/hendrycks/robustness
[12]https://scikit-learn.org/stable/modules/generated/sklearn.svm.SVR.html

We then estimate the properties given a trained regression model on the 500 ID-TEST architectures of DEEPNETS-1M and measure Kendall's Tau rank correlation with the ground truth test values. We repeat the experiment 5 times with different random seeds, i.e. different cross-validation folds. In Fig. 4, we show the average and standard deviation of correlation results across 5 runs. To train the graph convolutional network of Neural Predictor [80], we use the open source implementation[13] and train it on the 500 validation architectures from scratch for each property.

**Downstream results.** Next, we verify if higher correlations translate to downstream gains. We consider the clean accuracy on CIFAR-10 in this experiment as an example. We retrain the regression model on the graph embeddings of the combined ID-VAL and ID-TEST sets and generate 100k new architectures (similarly to how ID-TEST are generated) to pick the most accurate one according to the trained regression model. We train the chosen architecture from scratch following [19, 24, 32], i.e. with SGD for 600 epochs using cutout augmentation (C), an auxiliary classifier (A) and the drop path (D) regularization. In addition, we train the chosen architecture for just 50 epochs using the same hyperparameters we used to train our ID-VAL and ID-TEST architectures as in § 5.1. We train the chosen architecture 5 times using 5 random seeds and report an average and standard deviation of the final classification accuracy on the test set of CIFAR-10 (Table 9). We perform this experiment for GHN-1 and GHN-2 in the same way. Among the methods we compare in Table 9, our GHN-2-based search finds the most performant network if training is done for 50 epochs (without and with C, A and D) and finds a competitive network if training is done for 600 epochs with C, A and D.

Table 9: CIFAR-10 best architectures and their performance on the test set. C — cutout augmentation, A — auxiliary classifier, D — drop path regularization. The best result in each row is bolded.

| | **GHN-1** | **GHN-2** | **DARTS** [19] | **PDARTS** [32] |
|---|---|---|---|---|
| |  |  |  |  |
| # params (M) | 3.1 | 3.1 | 3.3 | 3.4 |
| 50 epochs | $92.61_{\pm 0.16}$ | $\mathbf{93.94}_{\pm 0.11}$ | $93.11_{\pm 0.09}$ | $92.95_{\pm 0.14}$ |
| 50 epochs + C,A,D [19] | $91.80_{\pm 0.14}$ | $\mathbf{95.24}_{\pm 0.14}$ | $94.50_{\pm 0.08}$ | $94.22_{\pm 0.06}$ |
| 600 epochs + C,A,D [19] | $95.90_{\pm 0.08}$ | $97.26_{\pm 0.09}$ | $97.17_{\pm 0.06}$ | $\mathbf{97.48}_{\pm 0.06}$ |

#### C.2.4 Comparing Neural Architectures

In addition to our experiments in § 5.2, we further evaluate representation power of GHNs by computing the distance between neural architectures in the graph embedding space.

**Experimental setup.** We compute the pairwise distance between the computational graphs $a_1$ and $a_2$ as the $\ell_2$ norm between their graph embedding $||\mathbf{h}_{a_1} - \mathbf{h}_{a_2}||$. We chose ResNet-50 as a reference network and compare its graph structure to the other three predefined architectures: ResNet-34, ResNet-50 without skip connections, and ViT. Among these, ViT is intuitively expected to have the largest $\ell_2$ distance, because it is mainly composed of non-convolutional operations. ResNet-50-No-Skip should be closer (in the $\ell_2$ sense) to ResNet-50 than ViT, because it is based on convolutions, but further away than ResNet-34, since it does not have skip connections.

**Additional baseline.** As a reference graph distance, we employ the Hungarian algorithm [43]. This algorithm computes the total assignment "cost" between two sets of nodes. It is used as an efficient approximation of the exact graph distance algorithms [106, 107], which are infeasible for graphs of the size we consider here.

**Qualitative results.** The $\ell_2$ distances computed based on GHN-2 align well with our initial hypothesis of the relative distances between the architectures as well as with the Hungarian distances (Table 10). In contrast, the baselines inappropriately embed ResNet-50 closer to ResNet-50-No-Skip

---

[13]https://github.com/ultmaster/neuralpredictor.pytorch

Table 10: Comparing ResNet-50 to the three predefined architectures using the GHNs trained on CIFAR-10 in terms of the $\ell_2$ distance between graph embeddings.

| RESNET-50 | RESNET-34 | RESNET-50-NO-SKIP | VIT |
|---|---|---|---|

| | RESNET-34 | RESNET-50-NO-SKIP | VIT |
|---|---|---|---|
| Hungarian | 118.0 | 134.0 | 207.5 |
| MLP | 0.4 | 0.2 | 1.0 |
| GHN-1 | 0.6 | 0.5 | 1.7 |
| GHN-2 | 1.0 | 1.3 | 3.1 |

than to ResNet-34. Thus, based on this simple evaluation, GHN-2 captures graph structures better than the baselines.

We further compare the architectures in the ID-TEST set and visualize the most similar and dissimilar ones using our trained models. Based on the visualizations in Fig. 12, MLP as expected is not able to capture graph structures, since it relies only on node features (the most similar architectures shown on the left are quite different). The difference between GHN-1 and GHN-2 is hard to understand qualitatively, so we compare them numerically below.

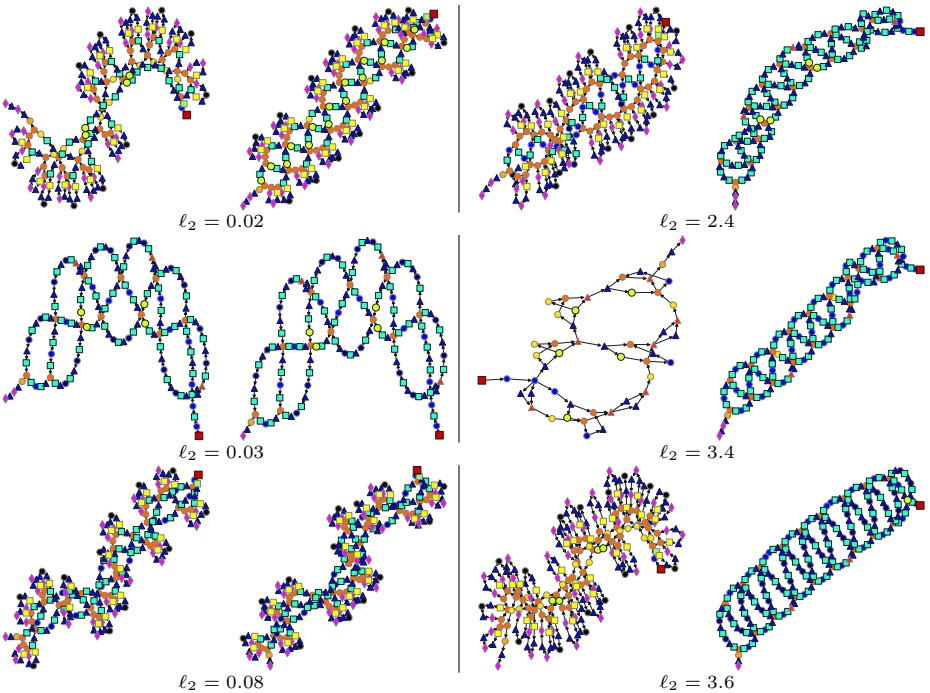

$\ell_2 = 0.02$ $\ell_2 = 2.4$

$\ell_2 = 0.03$ $\ell_2 = 3.4$

$\ell_2 = 0.08$ $\ell_2 = 3.6$

Figure 12: Most similar (left) and dissimilar (right) architectures (in terms of the $\ell_2$ distance) in the ID-TEST set based on the graph embeddings obtained by the MLP (top row), GHN-1 (middle row) and GHN-2 (bottom row) trained on CIFAR-10.

**Quantitative results.** To quantify the representation power of GHNs, we exploit the fact that, by design, the OOD architectures in our DEEPNETS-1M are more dissimilar from the ID architectures than the architectures within the ID distribution. So, a strong GHN should reflect that design principle and map the OOD architectures further from the ID ones in the embedding space. One way to measure the distance between two feature distributions, such as our graph embeddings, is the Fréchet distance (FD) [108]. We compute the FD between graph embeddings of 5000 training architectures and five test subsets (in the similar style as in [109–111]): ID-TEST and four OOD subsets (Table 11).

GHN-2 maps the ID-TEST architectures close to the training ones, while the OOD ones are further away, reflecting the underlying design characteristics of these subsets. On the other hand, both baselines inappropriately map the DEEP and DENSE architectures as close or even closer to the training ones than ID-TEST, despite the differences in their graph properties (Table 1, Fig. 6). This indicates GHN-2's stronger representation compared to the GHN-1 and MLP baselines.

Table 11: FD between test and training graph embeddings on CIFAR-10. The average assignment cost between two sets of nodes using the Hungarian algorithm is shown for reference.

| MODEL | ID-TEST | WIDE | DEEP | DENSE | BN-FREE |
|---|---|---|---|---|---|
| Hungarian | 208.7 | 199.9 | 365.6 | 340.7 | 193.1 |
| MLP | 0.50 | 1.04 | 0.37 | 0.39 | 1.10 |
| GHN-1 | 0.15 | 0.45 | 0.16 | 0.21 | 5.45 |
| GHN-2 | 0.30 | 0.80 | 1.11 | 0.59 | 3.64 |

Alternative ways to compare graphs include running expensive graph edit distance (GED) algorithms, infeasible for graphs of our size, or designing task-specific graph kernels [78, 112, 77, 106]. As a more efficient (and less accurate) variant of GED, we employ the Hungarian algorithm. It finds the optimal assignment between two sets of nodes, so its disadvantage is that it ignores the edges. It still can capture the distribution shifts between graphs that can be detected based on the number of nodes and their features, such as for the DEEP and DENSE subsets (Table 11). It does not differentiate TRAIN and BN-FREE, perhaps, due to the fact that a small portion of architectures in TRAIN does not have BN. Finally, the inability of the Hungarian algorithm to capture the difference between TRAIN and WIDE can be explained by the fact that we ignore the shape of parameters when running this algorithm.

## C.3 Analysis of Predicted Parameters

### C.3.1 Diversity

We analyze how much parameter prediction is sensitive to the input network architecture. For that purpose, we analyze the parameters of 1,000 evaluation architectures (ID-VAL and ID-TEST) of DEEPNETS-1M on CIFAR-10. We compare the diversity of the parameters trained with SGD from scratch (He's initialization+SGD) to the diversity of the parameters predicted by GHNs. To analyze the parameters, we consider all the parameters associated with an operation, which is in general a 4D tensor (i.e. out channels × in channels × height × width). As tensor shapes vary drastically across architectures and layers, it is challenging to find a large set of tensors of the same shape. We found that the shape of $128 \times 1 \times 3 \times 3$ is one of the most frequent ones: appearing 760 times across the evaluation architectures. So for a given method of obtaining parameters (i.e. He's initialization+SGD or GHN), we obtain a set of 760 tensors with $128 \times 1 \times 3 \times 3 = 1152$ values in each tensor, i.e. a $760 \times 1152$ matrix. For each pair of rows in this matrix, we compute the absolute cosine distance, which results in a $760 \times 760$ matrix with values in range [0,1]. We report the mean value of this matrix in Table 12.

**Results.** The parameters predicted by GHN-2 are more diverse than the ones predicted by GHN-1: average distance is 0.17 compared to 0.07 respectively (Table 12). The parameters predicted by MLPs are extremely similar to each other, since the graph structure is not exploited by MLPs. The cosine distance is not exactly zero for MLPs, because two different primitives (group convolution and dilated group convolution) can have the same $128 \times 1 \times 3 \times 3$ shape. The parameters predicted by GHN-2 are more similar (low cosine distance) to each other compared to He's initialization+SGD. Low cosine distances in case of GHNs indicate that GHNs "store" good parameters to some degree However, our GHN-2 seems to rely on storing the parameters less than GHN-1 and MLP. Instead, GHN-2 relies more on the input graph to predict more diverse parameters depending on the layer and architecture. So, GHN-2 is more sensitive to the input graph. We believe this is achieved by our enhancements, in particular virtual edges, that allow GHN-2 to better propagate information across nodes of the graph.

### C.3.2 Sparsity

We also analyze the sparsity of predicted parameters using the same 1,000 evaluation architectures. The sparsity can change drastically across the layers, so to fairly compare sparsities we consider

Table 12: Analysis of predicted parameters on CIFAR-10.

| Method of obtaining parameters | Average distance between parameter tensors | Average sparsity |
|---|---|---|
| He's init. + SGD | 0.98 | 33% |
| MLP | 0.01 | 16% |
| GHN-1 | 0.07 | 20% |
| GHN-2 | 0.17 | 39% |

the first layer only. We compute the sparsity of parameters $\mathbf{w}$ as the percentage of absolute values satisfying $|\mathbf{w}| < 0.05$. We report the average sparsity of all first-layer parameters in Table 12.

**Results.** The first-layer parameters predicted by GHN-2 are similar to the parameters trained by SGD in terms of sparsity: average sparsity is 39% compared to 33% respectively (Table 12). Higher sparsity of the parameters predicted by GHN-2 (39%) compared to the ones of GHN-1 (20%) may have been achieved due to the proposed parameter normalization method. The parameters predicted by GHN-2 are also more sparse than the ones obtained by SGD. Qualitatively, we found that GHN-2 predicts many values close to 0 in case of convolutional kernels 5×5 and larger, which is probably due to the bias towards more frequent 3×3 and 1×1 shapes during training. Mitigating this bias may improve GHN's performance.

### C.4 Training Speed of GHNs

Training GHN-2 with meta-batching takes 0.9 GPU hours per epoch on CIFAR-10 and around 7.4 GPU hours per epoch on ImageNet (Table 13). The training speed is mostly affected by the meta-batch size ($b_m$) and the sequential nature of the GatedGNN. Given that GHNs learn to model 1M architectures, the training is very efficient. The speed of training GHNs with can be further improved by better parallelization of the meta-batch. Note that GHNs need to be trained only once for a given image dataset. Afterwards, trained GHNs can be used to predict parameters for many arbitrary architectures in less than a second per architecture (see Table 4 in the main text).

Table 13: Times of training GHNs on NVIDIA V100-32GB using our code.

| MODEL | # GPUs | CIFAR-10 64 images/batch | | IMAGENET 256 images/batch | |
|---|---|---|---|---|---|
| | | sec/batch | hours/epoch | sec/batch | hours/epoch |
| Training a single ResNet-50 with SGD | 1 | 0.10 | 0.02 | 0.77 | 1.03 |
| MLP with meta-batch size $b_m = 1$ | 1 | 0.32 | 0.06 | 0.67 | 0.90 |
| GHN-2 with meta-batch size $b_m = 1$ | 1 | 1.16 | 0.23 | 1.54 | 2.06 |
| GHN-2 with meta-batch size $b_m = 8$ | 1 | 7.17 | 1.40 | out of GPU memory | |
| GHN-2 with meta-batch size $b_m = 8$ | 4 | 4.62 | 0.90 | 5.53 | 7.40 |

## D Additional Related Work

Besides the works discussed in § 6, our work is also loosely related to other parameter prediction methods [113, 98, 114], analysis of graph structure of neural networks [46], knowledge distillation from multiple teachers [115], compression methods [116] and optimization-based initialization [117–119]. Denil et al. [113] train a model that can predict a fraction of network parameters given other parameters requiring to retrain the model for each new architecture. Bertinetto et al. [98] train a model that predicts parameters given a new few-shot task similarly to [18, 96], and the model is also tied to a particular architecture. The HyperGAN [114] allows to generate an ensemble of trained parameters in a computationally efficient way, but as the aforementioned works is constrained to a particular architecture. Finally, MetaInit [117], GradInit [118] and Sylvester-based initialization [119] can initialize arbitrary networks by carefully optimizing their initial parameters, but due to the optimization loop they are generally more computationally expensive compared to predicting parameters using GHNs. Overall, these prior works did not formulate the task nor proposed the methods of predicting performant parameters for diverse and large-scale architectures as ours.

Finally, the construction of our DEEPNETS-1M is related to the works on network design spaces. Generating arbitrary architectures using a graph generative model, e.g. [120, 121, 46], can be one way to create the training dataset $\mathcal{F}$. Instead, we leverage and extend an existing DARTS framework [19] specializing on neural architectures to generate $\mathcal{F}$. More recent works [122] or other domains [123] can be considered in future work.

# E    Limitations

Our work makes a significant step towards reducing the computational burden of iterative optimization methods. However, it is limited in several aspects.

**Fixed dataset and objective.** Compared to GHNs, one of the crucial advantages of iterative optimizers such as SGD is that they can be easily used to train on new datasets and new loss functions. Future work can thus focus on fine-tuning GHNs on new datasets and objectives or conditioning GHNs on data and hyperparameters akin to SGD.

**Training speed.** Training GHN-2 on larger datasets and with a larger meta-batch ($b_m$) becomes slower. For example, on ImageNet with $b_m = 8$ it takes about 7.4 hours per epoch using $4\times$NVIDIA V100-32GB using our PyTorch implementation. So, training of our GHN-2 on ImageNet for 150 epochs took about 50 days. The slow training speed is mainly due to limited parallelization of the meta-batch (i.e. using $b_m$ GPUs should be faster) and the sequential nature of the GatedGNN.

**Fine-tuning predicted parameters.** While in § 5.3 we showed significant gains of using GHN-2 for initialization on two low-data tasks, we did not find such an initialization beneficial in the case of more data. In particular, we compare training curves of ResNet-50 on CIFAR-10 when starting from the predicted parameters versus from the random-based He's initialization [57]. For each initialization case, we tune hyperparameters such as an initial learning rate and weight decay. Despite ResNet-50 being an OOD architecture, GHN-2-based initialization helps the network to learn faster in the first few epochs compared to He's initialization (Fig. 13). However, after around 5 epochs, He's initialization starts to outperform GHN-2, diminishing its initial benefit. By fine-tuning the predicted parameters with Adam we could slightly improve GHN-2's results. However, the results are still worse than He's initialization in this experiment, which requires further investigation. Despite this limitation, GHN-2 significantly improves on GHN-1 in this experiment similarly to § 5.3.

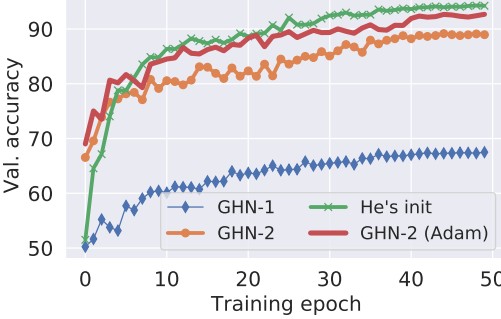

Figure 13: Training ResNet-50 on CIFAR-10 from random and GHN-based initializations.

# F    Societal Impact

One of the negative impacts of training networks using iterative optimization methods is the environmental footprint [11, 12]. Aiming at improving on the state-of-the-art, practitioners often spend tremendous computational resources, e.g. for manual/automatic hyperparameter and architecture search requiring rerunning the optimization. Our work can positively impact society by making a step toward predicting performant parameters in a resource-sustainable fashion. On the other hand, compared to random initialization strategies, in parameter prediction a single deterministic model could be used to initialize thousands or more downstream networks. All predicted parameters can inherit the same bias or expose any offensive or personally identifiable content present in a source dataset.