# OpenReview forum: "Parameter Prediction for Unseen Deep Architectures"
_NeurIPS.cc/2021/Conference — NeurIPS 2021 Poster_

### Official Review · Reviewer_mxL9 · 2021-07-18

**Rating:** 7
**Confidence:** 4

**Summary:**

This paper proposes and analyzes a method for generating trained weights already optimized for a given single task for a large class of diverse architectures using Graph hypernetworks. They introduce a new DEEPNETS-1M benchmark with in-distribution and out of distribution architectures. The work also proposed three main improvements to improve performance of their GHN: normalization of predicted parameters, enhanced long-range interactions in the GHN, and meta-batching of architectures.  They demonstrate that their method is able to produce weights that do surprisingly well on unseen and diverse networks in less than 1 second on CPU or GPU for CIFAR-10 and Imagenet. They demonstrate that even for large Resnet-50s which are out of the training distribution, they could achieve 60% accuracy on CIFAR10.

**Limitations And Societal Impact:**

The paper adequately addressed the limitations and potential negative societal impact of their work.

**Main Review:**

This paper analyzes the interesting task of directly predicting trained parameters for diverse feed-forward networks. This in itself is not novel since it has been done previously in many NAS works for the purposes for creating performance predictors as well works on targeting a specific class of networks. However, there is significant novelty in the scope of diverse network architectures that their method can predict reasonable weights for and this may be the first which does a focused analysis on accuracy performance of these predicted weights. The analysis of results on in distribution and out of distribution architectures is quite novel. Unfortunately in terms of practical significance, the practical use cases of using these weights to generate good initializations for fine tuning seems to currently significantly degrade performance. That in itself is still an interesting finding to invite further analysis.

The paper provides thorough analysis of their methods against the baseline of training from scratch for a very diverse set of architectures. The work also proposed three main improvements to improve performance: normalization of predicted parameters, enhanced long-range interactions in the GHN, and meta-batching of architectures. They provide thorough experiments ablating the effect of these changes.

Some of the results could benefit from further analysis. For example, it's interesting that the CIFAR-10 trained networks seem to generalize significantly better to Resnet than the Imagenet-trained networks.

I believe that this paper's main weakness is that it would be very useful to be able to compare the performance of the GHN with some stronger competitors. While there may be little or no work on predicting weights for out-of-distribution architectures, there has been significant NAS work on training supernets. A strong baseline then would be to train a supernet on some subset of the architecture space using a uniform prior and examine performance when directly deriving networks from those subnetworks.

In addition, this paper would greatly benefit from significant analysis of the weights that are generated. How sparse are the weights generated? Is there a correlation between more weights or less weights and model accuracy? For deep networks for example is it generating a few layers and a bunch of identity or skip layers? It would be interesting to visualize the activations.

Currently, it is also a bit difficult to judge the usability of this work since it lacks discussion of the training cost of the graph hypernetwork.


**Time Spent Reviewing:**

6

---

> ### Author Response · Authors · 2021-08-11
> **Response to Reviewer mxL9**
>
> We thank Reviewer **mxL9** for the constructive feedback. Please see below our responses to **mxL9**'s comments.
>
> > ... the practical use cases of using these weights to generate good initializations for fine tuning seems to currently significantly degrade performance.
>
> Please see **“Fine-tuning predicted parameters” in Common Response to All Reviewers - Part 1**.
>
> > it's interesting that the CIFAR-10 trained networks seem to generalize significantly better to Resnet than the Imagenet-trained networks.
>
> We speculate it might be partially due to overfitting of GHN-2 to the in-distribution architectures, which degrades generalization to OOD architectures such as ResNet-50. On ImageNet, there are more training iterations, so the GHN-2 observes more architectures. Assuming all sampled architectures are different, it is enough to have just 25 epochs for GHN-2 to observe all 1M architectures. On CIFAR-10, 175 epochs are required to observe all training architectures, so overfitting to the training architectures is less likely to occur. We empirically confirmed that GHN-2 trained on ImageNet for just 20 epochs actually has a better result (than the final GHN-2 checkpoint) when predicting parameters for ResNet-50: top-5 accuracy is 10.3% vs 5.0%. Earlier checkpoints of GHN-1 do not improve the 6.8% top-5 accuracy obtained using the final GHN-1 checkpoint (reported in Table 3 in the submitted paper).
>
> > A strong baseline then would be to train a supernet...
>
> We have a strong intuition about the expected results of the suggested experiment. We have considered OnceForAll (OFA) [92] as one of the stronger supernet-based baselines. There are two fundamental issues with OFA. First, to make it feasible to train, our training set of architectures must be significantly constrained in terms of the number of operators and the shapes of their associated weights. Such a constraint would defeat the purpose of creating our DeepNets-1M dataset. Second, OFA is not designed to generalize to large architectures. It’s the opposite. The training architecture is extremely large and all the derived ones must not be larger. So even if there is an implementation trick to make it possible to obtain the weights for larger architectures using OFA, such networks are expected to have performance close to random.
>
> > this paper would greatly benefit from significant analysis of the weights that are generated. How sparse are the weights generated?
>
> Please see **Table 3 in Common Response to All Reviewers - Part 2**.
>
> > Is there a correlation between more weights or less weights and model accuracy? For deep networks for example is it generating a few layers and a bunch of identity or skip layers?
>
> Please see Section C.2.2 and Fig. 12 in Supplementary Material, where we plot performance vs. various hyperparameter choices (network width, depth, number of nodes, and depth for networks without batch norm). Overall, the accuracy of predicted parameters of a network is mainly affected by how far the network is from the training distribution. For example, the predicted parameters of wider networks with more weights generally have lower accuracy than narrow networks, because during training GHNs observe more narrow networks.
>
> > It would be interesting to visualize the activations.
>
> Please see “activation” in Fig. 10 of the Supplementary Material. We plot the variance of activations for ResNet-50 on CIFAR-10 depending on the layer. ResNet-50 with the parameters predicted by GHN-2 has similar variances of activations to the ResNet-50 trained with SGD. In contrast, in case of GHN-1, the variance of activations grows with depth very fast making the activations very different from those of ResNet-50 trained with SGD.
>
> > it is also a bit difficult to judge the usability of this work since it lacks discussion of the training cost of the graph hypernetwork.
>
> Please see **Table 4 in Common Response to All Reviewers - Part 2**.

---

> > ### Comment · Reviewer_mxL9 · 2021-09-02
> > **Response to Authors**
> >
> > I would first like to thank the authors for the addition of these experiments in the low-data regime and for answering most of my questions. I have increased my score and overall I would recommend acceptance for this paper.
> >
> > If possible, I would still suggest you compare the distribution results with OFA even though as you said it doesn't have the same capabilities as your method.

---

### Official Review · Reviewer_YdDc · 2021-07-19

**Rating:** 6
**Confidence:** 3

**Summary:**

This paper propose a way to predict/generate the parameters of an image based deep network instead of training it via SGD. It's a thought-provoking paper and the authors appears to be very detailed in their implementation. The results are surprisingly good but not better than standard optimization algorithms.

**Ethics Review Area:**

["I don’t know"]

**Limitations And Societal Impact:**

Yes.

**Main Review:**

This paper propose a way to predict/generate the parameters of an image based deep network instead of training it via SGD. It's a thought-provoking paper and the authors appears to be very detailed in their implementation. The results are surprisingly good but not better than standard optimization algorithms.

Positives of the paper:
- The idea is novel and very thought provoking and different than standard way of approaching learning. The ability to accelerate learning is a great thing to do, given the time it takes to train large models.
- The paper is well written and clear. The authors have also shown the limitations of this approach which provides good context and discussion.

Negatives of the paper:
- The results doesn't surpass He-initialization with finetuning, which would've be nice and strong suggestions in favor this this approach.
- The authors suggested that it was 'natural' to be able to predict parameters from architecture. I think this isn't quite clear and it would be good for the author to outline more benefits to predicting the parameters. Given that the weights are good but not near SoTA, it's is naturally to ask why predict the parameters?
- The proposed method predicts parameters given architecture. How does this fit in with NAS?
- How sensitive are the changes of the parameters to network architecture (of the same parameter dimension), is it possible that the good weights are 'stored' in the GHN ?

Overall I think the paper is well written and novel and interesting, even though the results are not SoTA. The authors are advised to address the comments above for the next version of the paper.


**Time Spent Reviewing:**

2

---

> ### Author Response · Authors · 2021-08-11
> **Response to Reviewer YdDc**
>
> We thank Reviewer **YdDc** for the constructive feedback. Please see below our responses to **YdDc**'s comments.
>
> > The results doesn't surpass He-initialization with finetuning, which would've be nice and strong suggestions in favor this this approach.
>
> Please see **Tables 1 and 2 in Common Response to All Reviewers - Part 1**.
>
> > ... it would be good for the author to outline more benefits to predicting the parameters.
>
> We agree that the word ‘natural’ does not describe our motivation of predicting parameters. A better word is “beneficial”. We highlight the following benefits of predicting parameters:
>
> - Computational costs. For example, SGD requires $10^4$ more time on a GPU to achieve the same accuracy as using GHN-2 (see Table 3 in the submitted paper). While the accuracy of GHN-2 is still relatively low compared to training with SGD for longer, we expect more progress in the future in our parameter prediction task. Eventually, we expect bridging the gap between predicting parameters and training with SGD which will allow practitioners and researchers to rely less on computationally intensive iterative optimization. Once the gap is narrowed, the parameter prediction task may be made more complex to add conditioning on the (unseen) dataset/domain/etc.
>
> - Transfer learning in the low-data regime (see “Fine-tuning predicted parameters” in **Common Response to All Reviewers - Part 1**).
>
> - Efficient estimation of network accuracy, i.e. NAS (see **NAS results in Common Response to All Reviewers - Part 1**).
>
> - Efficient estimation of network properties besides the accuracy, so we can efficiently and effectively search for more fast, accurate and robust networks (see Section 5.2 in the paper and C.2.3 In Supplementary Material).
>
> - Other benefits that we have not discussed are, for example, network compression. For instance, instead of storing 25M parameters of ResNet-50 and other large models, we only need to store around 2M parameters of a GHN.
>
> > The proposed method predicts parameters given architecture. How does this fit in with NAS?
>
> Please see the **NAS results in Common Response to All Reviewers - Part 1**.
>
> > How sensitive are the changes of the parameters to network architecture (of the same parameter dimension), is it possible that the good weights are 'stored' in the GHN ?
>
> Please see **Table 3 in Common Response to All Reviewers - Part 2**.

---

### Official Review · Reviewer_KeMh · 2021-07-26

**Rating:** 7
**Confidence:** 3

**Summary:**

In this paper, the authors propose the task of parameter prediction with a new DeepNets-1M dataset. By using a improved Graph HyperNetwork, they can achieve reasonable performance for unseen and diverse networks without iterative optimization. The learned neural architecture representation also performs better than previous works.

**Limitations And Societal Impact:**

- Although the authors claim this parameters prediction task is more general than NAS, I think the most suitable application scenario of this work is still in NAS (given the performance of predicted parameters still have a huge gap with iterative optimized parameters). It would be interesting to see some results in NAS. I believe the network representation learned by GHN-2 should be powerful for NAS.
- Including some OOD architectures in the training set would be an interesting direction. It's great to see whether this framework can work for architectures with mixed ops (for example an architecture with both convolutions and multi-head attentions).
- It would be interesting to see whether this work can be applied to other tasks beyond image classification (e.g. object detection).

**Main Review:**

- The three components proposed by this paper significantly improve the original GHN performance. The authors also provide rich ablation studies to show the effectiveness of their modifications.
- The authors extend the network design space to include a lot of different ops. The proposed GHN-2 shows good generalization ability on OOD architectures.
- The neural architecture representation learned by GHN seems to be more powerful than previous works.

**Time Spent Reviewing:**

4

---

> ### Author Response · Authors · 2021-08-11
> **Response to Reviewer KeMh**
>
> We thank Reviewer **KeMh** for the constructive feedback. Please see below our responses to **KeMh**'s comments.
>
> > It would be interesting to see some results in NAS.
>
> Please see the **NAS results in Common Response to All Reviewers - Part 1**.
>
> > Whether this framework can work for architectures with mixed ops (for example an architecture with both convolutions and multi-head attentions)
>
> Yes, our framework can work for architectures with mixed ops in the same architecture. Such architectures are present in both the training and evaluation sets. All ops receive a $C \times H \times W$ tensor as input and return the tensor of the same shape, where $C$ is the number of channels, $H$ and $W$ are height and width respectively. So, both convolutions and multi-head attention can be applied. For example, when multi-head attention is used, the input is first reshaped to a 2D tensor ($C \times H W$) and the output is reshaped back to a 3D tensor ($C \times H \times W$), so that convolution can be applied.
>
> > It would be interesting to see whether this work can be applied to other tasks beyond image classification (e.g. object detection).
>
> Please see **Table 2 in Common Response to All Reviewers - Part 1**.

---

### Official Review · Reviewer_CZfY · 2021-08-03

**Rating:** 8
**Confidence:** 2

**Summary:**

This paper studied the problem of parameter prediction for deep neural networks. Especially, the authors focused on the task of predicting parameters for various network structures with a single hypernetwork. The method is based GHN and introduces several important modifications. To evaluate the performance of the proposed method, a standardized benchmark is introduced.  The improved method showed impressive performance on CIFAR-10 and ImageNet even with large neural networks. Besides, the method is also proved to be useful in architecture representation learning.



**Limitations And Societal Impact:**

Yes.
Refer to the main reviews.

**Main Review:**

Strengths:
1. The task of predicting the parameter of diverse architectures with a single hyper-network is novel and important. Though iterative optimization is the mainstream for learning deep networks, it should be encouraged to explore different directions.

2. The method is based on the existing work Graph-hypernetworks, while there are three key elements introduced to gain superior improvement. The modification is certainly well motivated which also provides impressive empirical performance.

3. The evaluation part of this paper is strict and extensive. I appreciate the efforts in providing the benchmarks for evaluating the generalization ability of the different methods which will certainly benefit the research in this area. The ablation study on different components is detailed and clear which makes the results convincing.

Questions:
1. One appealing property of the proposed method is the decreased computation burden. To this end, I suggest the authors provide some statistics about the learning procedure of the Graph-hypernetworks.

2. The experiments is conducted mainly in the image classification settings. I wonder whether the proposed method could apply to different data type, such as language and graph, or different architectures such as transformers.



**Time Spent Reviewing:**

4 hours

---

> ### Author Response · Authors · 2021-08-11
> **Response to Reviewer CZfY**
>
> We thank Reviewer **CZfY** for the constructive feedback. Please see below our responses to **CZfY**'s comments.
>
> > I suggest the authors provide some statistics about the learning procedure of the Graph-hypernetworks.
>
> Please see **Table 4 in Common Response to All Reviewers - Part 2** for the statistics.
>
> > I wonder whether the proposed method could apply to different data type, such as language and graph, or different architectures such as transformers.
>
> Other modalities (language, graphs) are not currently supported in our implementation. However, we do not see any fundamental limitations to train GHNs on other modalities, since the training algorithm and GHN’s architecture are not tied to data type. Extending GHNs to other modalities is interesting for future research.
>
> Regarding different architectures, our GHNs are able to predict parameters for Transformers. In particular, we predict parameters for Visual Transformers (ViT) that we use for image classification (see Tables 2 and 3 in the submitted paper and **Tables 1 and 2 in Common Response to All Reviewers - Part 1**).

---

> > ### Comment · Reviewer_CZfY · 2021-08-31
> > **Response to Rebuttal**
> >
> > Thanks for the responses. The rebuttal fixed all my concerns which is impressive. Hence I increase my score and vote for acceptance.

---

### Author Response · Authors · 2021-08-11
**Common Response to All Reviewers - Part 1**

We thank all reviewers for the constructive feedback and their enthusiasm regarding our work. We appreciate that the reviewers highlighted the strengths of our paper: “significant novelty” (**mxL9**), “novel and important task” (**CZfY**), “interesting findings'' (**mxL9**) and “thought-provoking paper” (**YdDc**). In this comment, we address common questions. This comment is split in two parts (Part 1 and Part 2) due to space limits. Other questions are addressed under the corresponding reviews.

# Fine-tuning predicted parameters
We first address the following comments by reviewers **YdDc**, **mxL9** and **KeMh** that concern fine-tuning predicted parameters as one of the practical use-cases of our GHNs:

- **YdDc**: *“surpassing He’s initialization with finetuning would've been nice and strongly suggest in favor of this approach”.*
- **mxL9**: *“the practical use cases of using these weights to generate good initializations for fine tuning seems to currently significantly degrade performance.”*
- **KeMh**: *“It would be interesting to see whether this work can be applied to other tasks beyond image classification (e.g. object detection)”*. A typical way to solve the object detection task is to fine-tune a network trained on a large source dataset (e.g. ImageNet) on less target object detection data. Therefore, we group this comment with the other two comments.

We address these comments by presenting two fine-tuning experiments in the low-data regime, where GHN-based initialization appears to be more promising than in the experiment we presented in Section 7 (Limitations). Low-data tasks are challenging and often tackled using transfer learning (e.g. see works [Raghu2019, Zhai2020]). To perform transfer learning, we predict the parameters using GHNs trained on ImageNet. We use ImageNet instead of CIFAR-10 as source data, since ImageNet has more data to leverage when transferring the predicted parameters to other tasks. We follow our fine-tuning experiment from Section 7 and compare the results to He’s initialization.

## Experiment 1

The first experiment is fine-tuning the predicted parameters on a small subset of CIFAR-10 training samples. In particular, we use 1,000 training samples (100 labels per class) instead of all 50,000. We fine-tune ResNet-50, Visual Transformer (ViT) and a 14-cell architecture based on the DARTS best cell [23]. The hyperparameters of fine-tuning were tuned on the validation set (200 samples held-out of the 1,000 training samples).

**Table 1. CIFAR-10 test results (image classification accuracies, %) when training on 1,000 samples. Average and standard deviation for 3 runs with different random seeds is shown. For each architecture, similar GHN-2-based and ImageNet init. results are bolded. The final epoch for ResNet/ViT is 100 and for DARTS is 250.**

| Method                                                                                 |  ResNet-50 | Visual Transformer (ViT) | DARTS best cell |
|----------------------------------------------------------------------------------------|:----------:|:------------------------:|:---------------:|
| He’s init. + training with SGD                                                         | 41.0 ± 0.4 |        33.2 ± 0.3        |    45.4 ± 0.4   |
| GHN-1 + fine-tuning with SGD                                                           | 46.6 ± 0.0 |        23.3 ± 0.1        |    49.2 ± 0.1   |
| GHN-2 + fine-tuning with SGD                                                           | **56.4** ± 0.1 |       **41.4** ± 0.6        |   **60.7** ± 0.3   |
| ImageNet init. (1k steps) + fine-tuning with SGD                         | 45.4 ± 0.3 |        **44.3** ± 0.1        |    **62.4** ± 0.3   |
| ImageNet init. (2.5k steps) + fine-tuning with SGD                        | **55.4** ± 0.2 |        50.4 ± 0.3        |    70.4 ± 0.2   |
| ImageNet init. (5k steps) + fine-tuning with SGD                         | 68.1 ± 0.2 |        57.4 ± 0.4        |    76.4 ± 0.2   |
| ImageNet init. (1 epoch = 10k steps) + fine-tuning with SGD                                        | 75.7 ± 0.6 |        61.9 ± 0.3        |    80.1 ± 0.1   |
| ImageNet init. (5 epochs) + fine-tuning with SGD                                       | 84.6 ± 0.2 |        70.2 ± 0.5        |    83.9 ± 0.1   |
| ImageNet init. (final epoch) + fine-tuning with SGD | 89.2 ± 0.2 |        74.5 ± 0.2        |    85.6 ± 0.2   |

The image classification results of fine-tuning the parameters predicted by our GHN-2 are ⪆10 percentage points better (in absolute terms) than fine-tuning the parameters predicted by GHN-1 or training the parameters initialized using He’s method (Table 1).

## Experiment 2

In the second experiment, we fine-tune the predicted parameters for the object detection task. We closely follow the “PyTorch Object Detection Finetuning Tutorial” [PyTorchTutorial] and train networks on the “Penn-Fudan Database for Pedestrian Detection and Segmentation”. The dataset contains 170 images and the task is to detect pedestrians. Since it is a small dataset, it is a good testbed for transfer learning. We consider the same three architectures as in *experiment #1*. Following [PyTorchTutorial], we replace the backbone of a Faster R-CNN with one of the three architectures. As in *experiment #1*, we compare He’s initialization to initializing based on the parameters predicted by GHNs trained on ImageNet.

**Table 2. Penn-Fudan object detection results (average precision at IoU=0.50).**

| Method                                                                                  |   ResNet-50   | Visual Transformer (ViT) | DARTS best cell |
|-----------------------------------------------------------------------------------------|:-------------------:|:------------------------:|:---------------:|
| He’s init. + training with SGD                                                          | 0.180 ± 0.012 | 0.145 ± 0.025            | 0.486 ± 0.035   |
| GHN-1 + fine-tuning with SGD                                                            | 0.450 ± 0.031 | 0.153 ± 0.017            | 0.468 ± 0.024   |
| GHN-2 + fine-tuning with SGD                                                            | **0.493** ± 0.024 | **0.322** ± 0.035            | **0.785** ± 0.032   |
| ImageNet init. (1k steps) + fine-tuning with SGD                                        | 0.302 ± 0.022 | 0.182 ± 0.046            | **0.814** ± 0.033   |
| ImageNet init. (2.5k steps) + fine-tuning with SGD                                      | **0.571** ± 0.056 | **0.322** ± 0.073           | **0.823** ± 0.022 |
| ImageNet init. (5k steps) + fine-tuning with SGD                                        | 0.674 ± 0.015 | **0.308** ± 0.024            | **0.822** ± 0.034   |
| ImageNet init. (1 epoch = 10k steps) + fine-tuning with SGD                             | 0.724 ± 0.020 | **0.365** ± 0.039            | **0.781** ± 0.046   |
| ImageNet init. (5 epochs) + fine-tuning with SGD                                        | 0.723 ± 0.045 | 0.391 ± 0.024            | 0.827 ± 0.053   |
| ImageNet init. (final epoch) + fine-tuning with SGD  | 0.876 ± 0.011 | 0.468 ± 0.023            | 0.881 ± 0.023   |

The object detection results of fine-tuning the parameters predicted using our GHN-2 are consistently better than both GHN-1 and He’s initialization (Table 2). The GHN-2 results are a factor of 1.5-2 improvement over He’s initialization for all the three architectures. Using GHN-1 for initialization provides relatively small or no gains, similar to *experiments #1*.

Overall, both experiments clearly demonstrate the practical value of predicting parameters using our GHN-2.

**Update**

In response to reviewer **mxL9**’s question about the results for ImageNet pre-training, we have included this additional comparison for both low data-regime experiments (see **ImageNet init.** in Tables 1 and 2). In *experiment #1*, initialization using GHN-2 leads to performance similar to 1k-2.5k steps of pretraining on ImageNet depending on the architecture, while in *experiment #2* GHN-2’s performance is similar to 1k-10k steps of pretraining with SGD.
In both experiments, pretraining on ImageNet for just 5 epochs (50k SGD steps) provides very strong transfer learning performance and the final ImageNet checkpoints are only slightly better. These results align with previous works, e.g. [Neyshabur2020]. Therefore, we expect that once the parameter prediction models reach the level of training with SGD for >=5 epochs on ImageNet, these parameter prediction models can provide significantly more benefit in low-data transfer learning.

**References**

[Raghu2019] Transfusion: Understanding Transfer Learning for Medical Imaging

[Zhai2020] A Large-scale Study of Representation Learning with the Visual Task Adaptation Benchmark

[PyTorchTutorial] TorchVision Object Detection Finetuning Tutorial https://pytorch.org/tutorials/intermediate/torchvision_tutorial.html

[Neyshabur2020] What is being transferred in transfer learning?

# NAS results

The reviewers **KeMh** and **YdDc** asked about NAS results.
We did NAS experiments following a standard NAS protocol [21, 23] and briefly discussed the NAS results in L288 of the submitted paper and in more detail in Supplementary Material in Section C.2.3. We ran NAS experiments on CIFAR-10 by choosing architectures based on GHNs (trained on CIFAR-10). Using the best architecture chosen based on our GHN-2, we obtained an image classification accuracy of 97.26% on CIFAR-10 (standard deviation is 0.09% for 5 runs). In contrast, the architecture chosen based on the baseline GHN-1 achieves only 95.90±0.08%. We compare these results to DARTS [23] and PDARTS [32] that follow the same NAS protocol (see the comparison in Table 7 in Supplementary Material). Our CIFAR-10 result of 97.26% is on par with leading NAS approaches (e.g. 97.17±0.06% of DARTS).

---

> ### Author Response · Authors · 2021-08-11
> **Common Response to All Reviewers - Part 2**
>
> # Analysis of predicted parameters
>
> The reviewers **YdDc** and **mxL9** are interested to see analysis of the predicted parameters:
>
> - **YdDc**: *“How sensitive are the changes of the parameters to network architecture (of the same parameter dimension)”*.
>
> - **mxL9**: *“How sparse are the weights generated?”*
>
> To address the question by **YdDc** about the sensitivity of parameters to the network architecture, we analyze the parameters of 1,000 evaluation architectures on CIFAR-10. We compare the diversity of the parameters trained with SGD from scratch (He’s initialization+SGD) to the diversity of the parameters predicted by GHNs. To analyze the parameters, we consider all the parameters associated with an operation, which is in general a 4D tensor (i.e. out channels × in channels × height × width). As tensor shapes vary drastically across architectures and layers, it is challenging to find a large set of tensors of the same shape. We found that the shape of 128×1×3×3 is one of the most frequent ones: appearing 760 times across the evaluation architectures. So for a given method of obtaining parameters (i.e. He’s initialization+SGD or GHN), we obtain a set of 760 tensors with 128×1×3×3 = 1152 values in each tensor, i.e. a 760×1152 matrix. For each pair of rows in this matrix, we compute the absolute cosine distance, which results in a 760x760 matrix with values in range [0,1]. We report the mean value of this matrix in Table 3.
>
> To address the question by **mxL9** regarding sparsity, we also analyze the parameters of 1,000 evaluation architectures. The sparsity can change drastically across the layers, so to fairly compare sparsities we consider the first layer only. We compute the sparsity of parameters **w** as the percentage of absolute values |**w**| < 0.05. We report the average sparsity of all first-layer parameters in Table 3.
>
> **Table 3. Analysis of predicted parameters on CIFAR-10.**
>
> | Method of obtaining parameters | Average distance between parameter tensors | Average sparsity |
> |--------------------------------|--------------------------------------------|------------------|
> | He’s init. + SGD        | 0.98                                       | 33%              |
> | MLP                            | 0.01                                       | 16%              |
> | GHN-1                          | 0.07                                       | 20%              |
> | GHN-2                          | 0.17                                       | 39%              |
>
> Regarding **YdDc**'s question:
>
> The parameters predicted by GHN-2 are more diverse than the ones predicted by GHN-1: average distance is 0.17 compared to 0.07 respectively (Table 3). The parameters predicted by MLPs are extremely similar to each other, since the graph structure is not exploited by MLPs. The cosine distance is not exactly zero for MLPs, because two different primitives (group convolution and dilated group convolution) can have the same 128×1×3×3 shape. The parameters predicted by GHN-2 are more similar (low cosine distance) to each other compared to He’s initialization+SGD. Low cosine distances in case of GHNs indicate that GHNs “store” good parameters to some degree as suggested by **YdDc**. However, our GHN-2 seems to rely on storing the parameters less than GHN-1 and MLP. Instead, GHN-2 relies more on the input graph to predict more diverse parameters depending on the layer and architecture. So, GHN-2 is more sensitive to the input graph. We believe this is achieved by our enhancements, in particular virtual edges, that allow GHN-2 to better propagate information across nodes of the graph.
>
> Regarding **mxL9**'s question:
>
> The first-layer parameters predicted by GHN-2 are similar to the parameters trained by SGD in terms of sparsity: average sparsity is 39% compared to 33% respectively (Table 3). Higher sparsity of the parameters predicted by GHN-2 (39%) compared to the ones of GHN-1 (20%) may have been achieved due to the proposed parameter normalization method. The parameters predicted by GHN-2 are also more sparse than the ones obtained by SGD. Qualitatively, we found that GHN-2 predicts many values close to 0 in case of convolutional kernels 5×5 and larger, which is probably due to the bias towards more frequent 3×3 and 1×1 shapes during training. Mitigating this bias may improve GHN’s performance.
>
> # Training cost of GHNs
>
> The reviewers **CZfY** and **mxL9** are interested in the computational requirements to train GHNs. We briefly mentioned the training speed in L368 (Section 7 - Limitations) of the submitted paper. Training GHN-2 with meta-batch size $b_m = 8$ takes 0.7 GPU hours per epoch on CIFAR-10 and around 7.8 GPU hours per epoch on ImageNet (see Table 4 below).
> The training speed is mostly affected by the meta-batch size ($b_m$) and the sequential nature of the GatedGNN. Given that GHNs learn to model 1M architectures, the training is very efficient. The speed of training GHNs with $b_m = 8$ can be further improved by better parallelization of the meta-batch. Note that GHNs need to be trained only once for a given image dataset. Afterwards, trained GHNs can be used to predict parameters for many arbitrary architectures in less than a second per architecture (see Table 3 in the submitted paper).
>
> **Table 4. Training cost (GPU hours per epoch) of GHNs using NVIDIA V100-32GB.**
>
> | GHN variant                          | CIFAR-10               | ImageNet                 |
> |--------------------------------------|------------------------|--------------------------|
> |                                      | 1 GPU, 64 images/batch | 4 GPUs, 256 images/batch |
> | Training a single ResNet-50 with SGD | 0.03                   | 0.83                     |
> | MLP with meta-batch size $b_m = 1$     | 0.04                   | 1.25                     |
> | GHN with meta-batch size $b_m = 1$     | 0.13                   | 2.08                     |
> | GHN with meta-batch size $b_m = 8$    | 0.70                   | 7.75                     |

---

> ### Comment · Reviewer_mxL9 · 2021-08-21
> **Fine-tuning Predicted Parameters**
>
> I would first like to thank the authors for the addition of these experiments in the low-data regime. It seems to improve performance and I am still fully considering your rebuttal. If you have the time, since I are not familiar with the difficulty of this tasks in these specific low data regimes could you include the accuracy with Imagenet pretraining for a few epochs? It would not be a fair comparison to your method obviously, but would help us understand the upper bound for performance in this low-data regime.

---

> > ### Author Response · Authors · 2021-08-25
> > **Update for Tables 1 and 2**
> >
> > We thank Reviewer **mxL9** for the follow-up question. Please see our updated results in Tables 1 and 2 and the related discussion in the paragraph under **Update**.

---

### Author Response · Authors · 2021-09-21
**Code release note**

We first thank the reviewers for the additional discussion and increasing the scores.

We have cleaned up our code and confirmed the reproducibility of our results. We are planning to release the code soon, once the usability testing is complete.

As a toy working example of using our code, below is the code to predict the parameters for ResNet-50 using PyTorch.

```
from ghn.ghn import GHN2
import torchvision.models as models

ghn = GHN2('imagenet')          # load our GHN-2 trained on ImageNet
model = models.resnet50()       # ResNet-50 or any other torchvision model
model = ghn(model)              # predict parameters in < 1 second on GPU/CPU
# That's it! The model can be now evaluated/fine-tuned.
```

---

### Decision · Program_Chairs · 2021-09-27

**Decision:**

Accept (Poster)

**Comment:**

This paper presents a technique for prediction of parameters in deep architectures. After the initial reviews, the reviewers were concerned about 1) performance in the low-data regime 2) training cost of GHNs 3) relevance/utility for NAS application 4) improvement via finetuning.
The authors clarified 1) and 4) well. 2) is high (as one would expect) but still reasonable, considering that predictions for new architectures can be quickly generated after initial training and that practitioners often adopt even more expensive approaches, such as OFA. The advantage of the OFA framework is that each architecture can be trained in parallel on a cluster, but then again, only a finite set of architectures can be trained this way, and the cost increases linearly with the number of architectures. 3) was briefly discussed in the paper and is relatively strong for the DARTS search space on CIFAR-10, although a comparison to standard/broader NAS benchmarks would have made a stronger point here.